# Learning General Representation of 12-Lead Electrocardiogram With a Joint-Embedding Predictive Architecture

## Abstract

Electrocardiogram (ECG) captures the heart's electrical signals, offering valuable information for diagnosing cardiac conditions. However, the scarcity of labeled data makes it challenging to fully leverage supervised learning in medical domain. Self-supervised learning (SSL) offers a promising solution, enabling models to learn from unlabeled data and uncover meaningful patterns. In this paper, we show that masked modeling in the latent space can be a powerful alternative to existing self-supervised methods in the ECG domain. We introduce ECG-JEPA, a SSL model for 12-lead ECG analysis that learns semantic representations of ECG data by predicting in the hidden latent space, bypassing the need to reconstruct raw signals. This approach offers several advantages in the ECG domain: (1) it avoids producing unnecessary details, such as noise, which is common in ECG; and (2) it addresses the limitations of naïve L2 loss between raw signals. Another key contribution is the introduction of Cross-Pattern Attention (CroPA), a specialized masked attention mechanism tailored for 12-lead ECG data. ECG-JEPA is trained on the union of several open ECG datasets, totaling approximately 180,000 samples, and achieves state-of-the-art performance in various downstream tasks including ECG classification and feature prediction.

## 1 Introduction

Electrocardiography is a non-invasive method to measure the electrical activity of the heart over time, serving as a crucial tool for diagnosing various cardiac conditions. While numerous supervised methods have been developed to detect heart diseases using ECG data (Hannun et al., 2019; Ribeiro et al., 2020; Siontis et al., 2021), these models often face significant performance degradation when applied to data distributions different from those on which they were trained. This challenge points to the need for more flexible approaches that can learn robust, transferable representations from ECG data.

Self-supervised learning (SSL) offers an alternative approach by learning general representations in diverse domains, such as natural language processing (NLP) (Devlin et al., 2019; Brown et al., 2020; Touvron et al., 2023), computer vision (CV) (Chen et al., 2020; He et al., 2022; Assran et al., 2023), and video analysis (Tong et al., 2022; Bardes et al., 2024). Despite this promise, the application of SSL to ECG data presents unique challenges. For instance, data augmentation, which is essential in many SSL architectures, is more complex for ECG than for computer vision data. Simple transformations like rotation, scaling, and flipping, effective in CV, can distort the physiological meaning of ECG signals. Additionally, ECG recordings often contain artifacts and noise, which may cause autoencoder-based SSL models to struggle with reconstructing raw signals. These architectures may also miss visually subtle but diagnostically critical features, such as P-waves and T-waves, which are imperative for diagnosing certain cardiac conditions.

In this work, we propose ECG Joint-Embedding Predictive Architecture (ECG-JEPA) tailored for 12-lead ECG data, effectively addressing the aforementioned challenges. ECG-JEPA utilizes a transformer architecture to capture the semantic meaning of the ECG. By masking several patches of the ECG, ECG-JEPA predicts abstract representations of the missing segments, indicating a high-level understanding of the data. Additionally, we develop a novel masked-attention for multi-lead ECG data, which we call Cross-Pattern

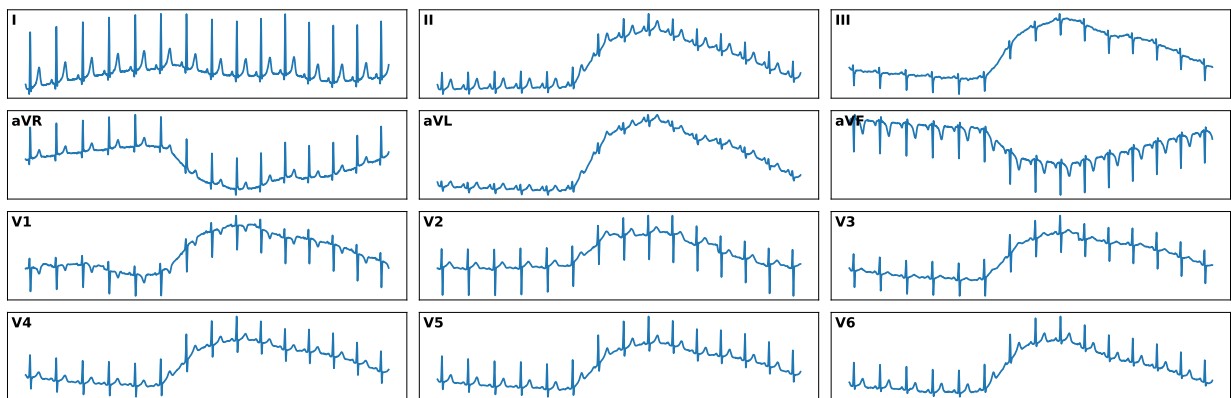

Figure 1: Example of 12-lead ECG signals from CODE-15 (Ribeiro et al., 2020) dataset.

Attention (CroPA). CroPA incorporates clinical knowledge into the model as an inductive bias, guiding it to focus on clinically relevant patterns and relationships across leads.

Our extensive empirical empirical evaluations reveals the following characteristics:

- ECG-JEPA achieves notable improvements in linear evaluation and fine-tuning on classification tasks compared to existing SSL methods without hand-crafted augmentations (Sections 5.1, 5.6).

- CroPA introduces a specialized masked attention mechanism, allowing the model to focus on clinically relevant information in multi-lead ECG data, resulting in improved downstream task performance (Section 5.7).

- ECG-JEPA can also recover important ECG features, including heart rate and QRS duration, which are classical indicators used in ECG evaluation. This is the first work to demonstrate that learned representations can effectively recover ECG features (Section 5.4).

- ECG-JEPA is highly scalable, allowing efficient training on large datasets. For instance, ECG-JEPA is trained for only 100 epochs, yet outperforms other ECG SSL models on most downstream tasks, taking approximately 22 hours on a single RTX 3090 GPU (Figure 3).

In summary, ECG-JEPA introduces a robust SSL framework for 12-lead ECG analysis, overcoming traditional SSL limitations with clinically inspired design elements, scalable architecture, and demonstrated effectiveness on a wide range of tasks.

## 2 Background

Self-Supervised Learning (SSL) facilitates learning abstract representations from input data without the need for labeled data, which is particularly beneficial in medical domains where labeled data is scarce and costly to obtain. SSL leverages inherent data patterns to learn useful representations, allowing models to adapt to various downstream tasks with greater robustness to data imbalances (Liu et al., 2022). We begin in Section 2.1 with an overview of the ECG and its key features, highlighting the critical characteristics essential for understanding ECG data. In Sections 2.2 and 2.3, we briefly explain key SSL techniques and their specific applications to ECG, respectively.

### 2.1 Electrocardiogram (ECG)

Electrocardiography is a non-invasive diagnostic method that records the heart's electrical activity over time using electrodes placed on the skin. The result of this recording is called an electrocardiogram (ECG), which

visually represents the electrical activity of the heart as a waveform. The standard 12-lead ECG captures electrical activity of the heart from multiple angles. These 12 leads are categorized into limb leads (I, II, III), augmented limb leads (aVR, aVL, aVF), and chest leads (V1-V6). Each lead provides unique information about the heart's electrical activity, offering a comprehensive view that aids in diagnosing various cardiac conditions. Refer to Figure 1 for an illustration of 12-lead ECG.

ECG features are specific characteristics of ECG signals that are critical for summarizing the overall signal. These features play an essential role in monitoring a patient's health status and are instrumental in the application of statistical machine learning models for diagnosing heart diseases. Key ECG features include heart rate, QRS duration, PR interval, QT interval, and ST segment. These features are identified by measuring specific time intervals or amplitude levels in the ECG waveform. For instance, heart rate is calculated using the formula $1000 \times (60/\text{RR interval})$ in beats per minute (bpm), where the RR interval is measured in milliseconds (ms). Refer to Figure 2 for a visual representation of these features.

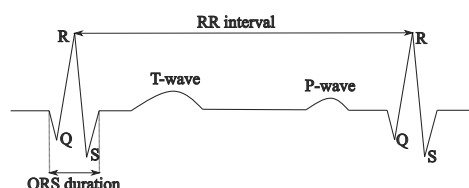

Figure 2: Key ECG features.

In this work, we use only 8 leads (I, II, V1-V6) as the remaining 4 leads (III, aVR, aVL, aVF) can be derived from linear combinations of the 8 leads following the *Einthoven's law* (Thaler, 2021):

$$\text{III} = \text{II} - \text{I}, \qquad \text{aVR} = -(\text{I} + \text{II})/2, \qquad \text{aVL} = (\text{I} - \text{II})/2, \qquad \text{aVF} = (\text{II} - \text{I})/2.$$

This choice maintains the necessary diagnostic information while optimizing computational efficiency. A performance comparison between the 8-lead model and the 12-lead model is provided in Appendix B.2, demonstrating that the 8-lead model achieves comparable results with reduced computational requirements.

## 2.2 Self-Supervised Learning Architectures

Self-supervised learning can be broadly categorized into contrastive and non-contrastive methods. Non-contrastive methods can be further divided into generative and non-generative architectures. See Balestriero et al. (2023) for a broader introduction to SSL.

In *contrastive learning*, the model is encouraged to produce similar representations for semantically related inputs $x'$ and $x''$, while pushing apart the representations of unrelated inputs $x'$ and $y'$. *SimCLR* (Chen et al., 2020) is one of the most popular contrastive methods, using two different augmentations of a single input $x$ to form semantically similar pairs $x'$ and $x''$.

Beyond contrastive methods, *generative architectures* have been particularly successful in recent large language models (Devlin et al., 2019; Brown et al., 2020; Touvron et al., 2023) and in computer vision (He et al., 2022). Generative architectures typically involve reconstructing a sample $x$ from its degraded version $x'$, employing either encoder-decoder frameworks or other paradigms like decoder-only or encoder-only models. The premise is that reconstructing clean data from a corrupted version reflects the model's deep understanding of the underlying data structure. In encoder-decoder frameworks, the encoder maps the perturbed input $x'$ into a latent representation, which the decoder then uses to reconstruct the original input $x$ (Vincent et al., 2008). Recently, Balestriero & LeCun (2024) observed that generative architectures prioritize learning principal subspaces of the data, which may limit their capacity to capture semantic representations for perceptual tasks.

As an alternative, *non-generative methods* have shown promise across domains, including computer vision (Grill et al., 2020; Bardes et al., 2022; Chen & He, 2020; Assran et al., 2023) and video analysis (Bardes et al., 2024). Among these, the Joint-Embedding Predictive Architecture (JEPA) (LeCun, 2022) processes an input pair $x$ and its corrupted versions $x'$ to obtain representations $z$ and $z'$ through encoders. Unlike generative architectures that make predictions in the input space, JEPA performs prediction in the latent

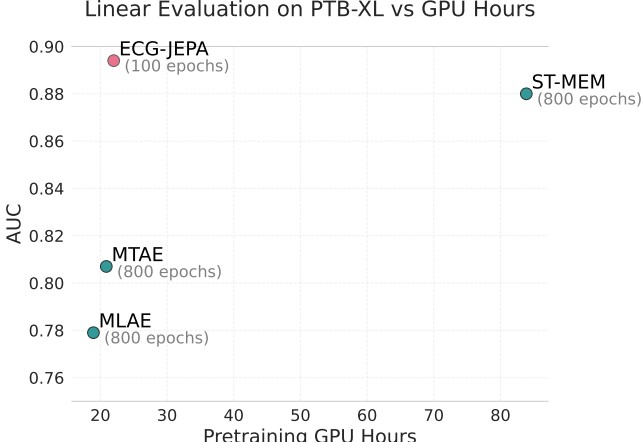

Figure 3: Linear evaluation on PTB-XL multi-class. ECG-JEPA makes predictions in the hidden representation space, while other methods are masked-autoencoder based, reconstructing raw signals.

space by reconstructing $z$ from $z'$. This approach effectively avoids the challenge of predicting unpredictable details, a common issue in biological signals.

## 2.3 Related Works

Several studies have focused on capturing semantically meaningful representations of 12-lead ECG data. *Contrastive Multi-segment Coding (CMSC)* (Kiyasseh et al., 2021) splits an ECG into two segments, encouraging similar representations for compatible segments while simultaneously separating incompatible ones. *Contrastive Predictive Coding (CPC)* (van den Oord et al., 2019), as applied in Mehari & Strodthoff (2022), predicts future ECG representations in a contrastive manner. Oh et al. (2022) combines Wav2Vec 2.0 (Baevski et al., 2020) with CMSC to capture both local and global features of ECG signals. More recently, many models (Zhang et al., 2022; Hu et al., 2023; Yang et al., 2022; Wang et al., 2023; Na et al., 2024) have adopted a masked autoencoder architecture, each employing its own unique masking strategy. Lastly, McKeen et al. (2024) utilizes the architecture suggested by Oh et al. (2022) to pretrain on large datasets (2.5 million samples) that include both open and private data.

## 3 Methodology

ECG-JEPA is trained by predicting masked patches of ECGs in the hidden representation space, using a partial view of the input to infer the missing parts. The proposed architecture utilizes a student-teacher framework, as illustrated in Figure 4. We subdivide the multi-channel ECG into non-overlapping patches and sample a subset of these patches for masking.

While our model is trained to predict in the representation space, learning by reconstructicting the raw signals can be particularly challenging in the ECG domain due to the prevalence of noise. Instead, our model predicts the masked patches in the hidden representation space, where this challenge can be effectively addressed.

Figure 3 illustrates the advantages of reconstructing in the hidden representation space by comparing the linear evaluation results of ECG-JEPA and other masked autoencoder-based models on the PTB-XL multi-class task. Blue dots represent masked autoencoder-based models. ECG-JEPA achieves superior performance with only 100 epochs of training.

Our approach inherently accounts for the presence of noise in biological signals, as the model is trained on raw ECG signals without any preprocessing or noise removal techniques. This design choice ensures that the model is trained on real-world noisy ECG samples (see Figure 1), enabling it to process such signals effectively, even when noise from sources like patient movement or electrical interference is present.

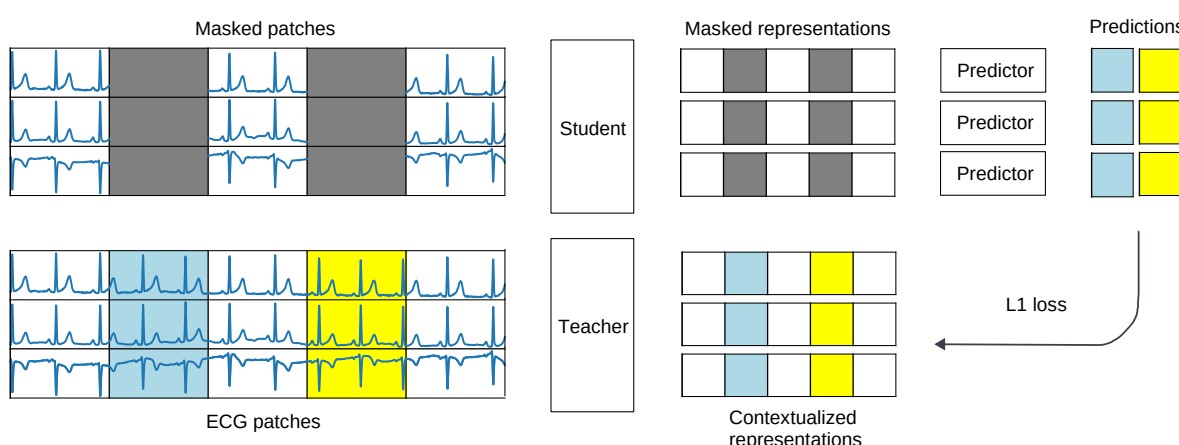

Figure 4: ECG-JEPA training overview. For illustration, we use $C = 3$ channels, $N = 5$ subintervals with $I_{vis} = \{1, 3, 5\}$, representing visible intervals and $I_{msk} = \{2, 4\}$, representing masked intervals.

## 3.1 Patch Masking

Let $x \in \mathbb{R}^{C \times T}$ represent a multi-lead ECG of length $T$ with $C$ channels. We divide the interval $[0, T)$ into $N$ non-overlapping subintervals of length $t$. Each subinterval in each channel constitutes a patch $x_{c,i} \in \mathbb{R}^t$ of $x$, resulting in $C \times N$ patches $\{x_{c,i}\}_{c \in [C], i \in [N]}$, where $[N]$ is the set of integers $\{1, 2, \ldots, N\}$.

The masking strategy in multi-lead ECG must be carefully chosen because patches in different leads at the same temporal position are highly correlated (Thaler, 2021), potentially making the prediction task too easy. To address this, we mask all patches across different leads in the same temporal space. With this in mind, we employ two masking strategies: *random masking* and *multi-block masking*.

In random masking, we randomly select a percentage of subintervals to mask, while in multi-block masking, we select multiple consecutive subintervals to mask. Note that we allow these consecutive subintervals to overlap, which requires the model to predict much longer sequences of representations. To evaluate the effectiveness of ECG-JEPA, we use both strategies, with a random masking ratio of $(0.6, 0.7)$ and a multi-block masking ratio of $(0.175, 0.225)$ at a frequency of 4 (see Appendix B.1 for an ablation study on varying masking ratios). For either masking strategy, the masking indices are denoted as $I_{msk} \subset [N]$, and the visible indices as $I_{vis}$, such that $[N] = I_{msk} \cup I_{vis}$. The unmasked patches $\{x_{c,i}\}_{c \in [C], i \in I_{vis}}$ serve as contextual input for the student networks, while the masked patches $\{x_{c,i}\}_{c \in [C], i \in I_{msk}}$ are the targets to predict in the representation space.

The patches $\{x_{c,i}\}_{c \in [C], i \in [N]}$ are converted into a sequence of token vectors $\{x_{c,i}^{\mathrm{tkn}}\}_{c \in [C], i \in [N]}$ of dimension $D$ using a linear layer, and augmented with positional embeddings. For simplicity, we continue to refer to the token vectors as $x_{c,i} \in \mathbb{R}^D$ with a slight abuse of notation. We employ the conventional 2-dimensional sinusoidal positional embeddings for the student and teacher networks, while 1-dimensional sinusoidal positional embeddings are used for the predictor network.

## 3.2 Teacher, Student, and Predictor

ECG-JEPA is built upon three key components: the teacher network, the student network, and the predictor network, each playing a distinct role in the model's learning process. The teacher and student networks are based on standard transformer architectures, while the predictor network, a smaller transformer, operates on single-channel representations. Despite operating on single channels, the predictor effectively encodes information from all leads, leveraging the self-attention mechanism to integrate contextual dependencies.

The teacher network handles the entire $C \times N$ patches $\{x_{c,i}\}_{c \in [C], i \in [N]}$, generating fully contextualized representations $\{z_{c,i}\}_{c \in [C], i \in [N]}$. The student network, however, processes only $C \times Q$ visible (unmasked) patches

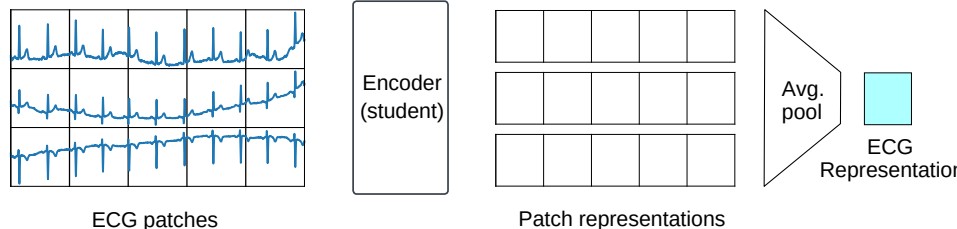

Figure 5: Patch-level representations are averaged to yield the ECG representation vector (colored in cyan).

$\{x_{c,i}\}_{c\in[C],i\in I_{vis}}$, where $Q = |I_{vis}|$ represents the number of visible time intervals. The representations $\{z_{c,i}^{\text{std}}\}_{c\in[C],i\in I_{vis}}$ from the student are then concatenated with $C\times(N-Q)$ (learnable) mask tokens $z_{msk} \in \mathbb{R}^D$, resulting in $C \times N$ representations. Subsequently, each lead's representations $\{z_{c,i}^{\text{std}}\}_{i\in I_{vis}} \cup \{z_{msk}, \ldots, z_{msk}\}$ are passed to the predictor, generating the predictions $\{\widehat{z_{c,i}}\}_{i\in[N]}$.

Finally, the objective function of ECG-JEPA is defined as the L1 distance between the predicted representations for the masked patches and their corresponding teacher-generated representations. Formally,

$$\mathcal{L} = \sum_{c\in[C]} \frac{1}{|I_{msk}|} \sum_{i\in I_{msk}} \|\widehat{z_{c,i}} - z_{c,i}\|_1$$

The main challenge in the student-teacher framework—or, more generally, in any joint-embedding architecture—is *model collapse*, where both encoders produce constant outputs regardless of their inputs, thereby minimizing the loss function. A common approach to prevent collapse is to update the teacher network's weights using an exponential moving average (EMA) of the student network's weights, which we adopt in our model. The details of EMA are provided in Appendix D.

### 3.3 ECG Representation

After training, only the student network is used as the encoder. The encoder's outputs are average-pooled to produce the final ECG representation, which serves as the feature vector for downstream tasks. The dimension of this latent representation vector matches the encoder's token dimension, which is set to $D = 768$ in our case. See Figure 5 for an illustration.

### 3.4 Cross-Pattern Attention (CroPA)

Interpreting a 12-lead ECG involves analyzing individual leads as well as comparing signals across multiple leads to enhance diagnostic accuracy. Multi-lead comparison not only helps distinguish artifacts from true abnormalities but also provides a more comprehensive assessment of cardiac activity. In Thaler (2021), for instance, the following quotes illustrate this principle:

> " ... The diagnosis (of posterior infarction) must therefore be made by looking for reciprocal changes in the anterior leads, for example, a tall R wave in leads V1, V2, or V3."

> "... ST-segment depression of at least 1 mm in leads V1-V3 if deep S waves are present is strongly suggestive of an evolving infarction."

Motivated by these observations, we introduce Cross-Pattern Attention (CroPA), a masked self-attention mechanism that imposes an inductive bias by restricting attention to clinically relevant patches. Specifically, a patch $x_{c,i}$ attends to another patch $x_{c',i'}$ if and only if either (1) they belong to the same lead ($c = c'$), or (2) they are in the same temporal space ($i = i'$) (see Section 3.2 for notations).

This design aligns with how ECG signals are clinically interpreted, where intra-lead and temporally adjacent signals are most relevant. By incorporating this inductive bias, CroPA focuses on relevant intra-lead relationships, reducing interference from unrelated signals across other channels and temporal spaces. Unlike standard self-attention, which treats all patches equally, CroPA adopts a structured approach that mirrors the clinical interpretation process, leading to improved performance on downstream tasks as demonstrated in Section 5.7.

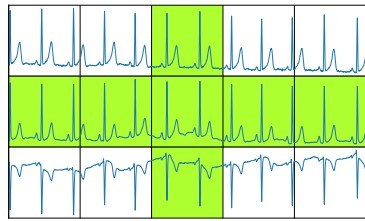

Figure 6: Cross-Pattern Attention.

## 4 Experimental Settings

In all experiments, 10-second multi-lead ECG signals were resampled to 250 Hz, yielding $T = 2500$ time points. We divided the interval $[0, T)$ into $N = 50$ non-overlapping subintervals, each of length $t = 50$. The model was trained for 100 epochs without data augmentation or noise removal preprocessing, and the final checkpoint was used for downstream tasks. Additional experimental details are provided in Appendix C.

### 4.1 Pretraining Datasets

Training SSL models with large datasets is crucial for developing generalized representations. However, most previous works have used relatively small datasets, with the exception of Na et al. (2024), where an SSL model was trained with a large number of 12-lead ECGs. Following Na et al. (2024), we use the *Chapman* (Zheng et al., 2020b), *Ningbo* (Zheng et al., 2020a), and *CODE-15* (Chen et al., 2019) datasets for pretraining ECG-JEPA. The Chapman and Ningbo datasets collectively consist of 45,152 10-second 12-lead ECGs at 500 Hz. CODE-15 includes 345,779 12-lead ECGs from 233,770 patients at 400 Hz, with 143,328 being 10-second recordings. After excluding recordings with missing values, we have 43,240 ECGs from Chapman and Ningbo and 130,900 ECGs from CODE-15.

### 4.2 Downstream Datasets

To evaluate the performance of ECG-JEPA on downstream tasks, we use the *PTB-XL* (Wagner et al., 2020) and *CPSC2018* (Liu et al., 2018) datasets.

*PTB-XL* provides three categories of labels for classification: *diagnostic labels*, *rhythm labels*, and *form labels*. Diagnostic labels correspond to specific conditions or pathologies identified in the ECG, rhythm labels describe the heart's rhythm or rate as observed in the ECG, and form labels capture morphological characteristics such as waveform shape and structure. In our experiments, we primarily focus on diagnostic labels, as they serve as the standard benchmark for ECG classification. Additionally, we visualize the learned representations of rhythm labels in Section 6.

The *PTB-XL diagnostic category* contains 21,837 12-lead ECG recordings sampled at 500 Hz, annotated with 71 diagnostic statements, which are grouped into five diagnostic superclasses. For our experiments, we use these superclass labels. The *PTB-XL rhythm category* consists of 16,687 recordings labeled into 12 rhythm classes. Unless otherwise specified, we refer to the *PTB-XL diagnostic category* when mentioning *PTB-XL* throughout this paper.

The *CPSC2018* dataset comprises 6,877 12-lead ECG recordings annotated with nine cardiac conditions. Both datasets follow a multi-label classification scheme, meaning each recording can be assigned multiple labels.

Further details on dataset characteristics are provided in Appendix C.1.

Although most experiments are conducted on *PTB-XL* and *CPSC2018*, we also performed a supplementary experiment using the Georgia 12-lead ECG Challenge (G12EC) Database (see Appendix A.4).

### 4.3  Architecture

Our model employs transformer encoder architectures for the student, teacher, and predictor networks. Both the teacher and student networks consist of 12 layers with 16 attention heads and a hidden dimension of 768. The predictor network, designed as a smaller transformer encoder, comprises 6 layers with 12 attention heads and a hidden dimension of 384. While the teacher and student networks process the multi-lead ECG data holistically, the predictor operates on each lead independently to reconstruct the masked representations. Importantly, this does not imply that the predictor relies solely on single-lead information for the reconstruction task; due to the self-attention mechanism, the input representations for each lead still encapsulate information from all leads.

### 4.4  Downstream Tasks

We conduct extensive experiments to show that ECG-JEPA effectively captures semantic representations. Its performance is evaluated on classification tasks using linear probing and fine-tuning. Furthermore, we assess its capability in low-shot learning settings, as well as under reduced-lead conditions where the downstream dataset is limited to single or two leads. Reduced-lead configurations are common in clinical practice, especially in scenarios like wearable devices or remote monitoring, where using the full 12-lead ECG setup is impractical.

To validate the expressiveness of the learned representations, we predict key ECG features such as heart rate and QRS duration. Notably, this work is the first to show that these learned representations can recover a variety of ECG features. The ability to predict these features highlights the informativeness of the representations and their potential to capture clinically relevant characteristics, which is crucial for reliable ECG analysis.

ECG datasets, such as *PTB-XL* and *CPSC2018*, often include multiple simultaneous labels for a single recording, making them multi-label tasks. However, many prior studies have simplified this into a multi-class classification problem by excluding samples having more than one labels. To ensure a fair comparison, we pretrain competing methods using publicly available code and evaluate them on the multi-label classification task. In cases where the code is unavailable, we will convert our task into a multi-class problem to align with the reported performance in the literature.

## 5  Experiments

In this section, we evaluate the performance of the learned representations across various downstream tasks to demonstrate their generalizability and ability to capture essential ECG features. ECG-JEPA is compared against several state-of-the-art self-supervised learning (SSL) methods.

For classification tasks, we use AUC (Area Under the ROC Curve) and F1 scores as evaluation metrics. AUC provides a comprehensive measure of discriminative ability by considering performance across all classification thresholds, making it more robust to variations in decision boundaries. In contrast, the F1 score balances precision and recall at a fixed threshold, offering insights into the model's performance when a specific decision boundary is chosen.

In multi-label classification, we compute AUC by averaging the scores from binary classification for each label, while for multi-class classification, AUC is calculated using the one-vs-rest approach. For both tasks, F1 scores are macro-averaged across all classes to ensure equal weighting of each class in the final score.

In most cases, ECG-JEPA consistently outperforms other SSL methods that rely on hand-crafted augmentations, highlighting its effectiveness in learning generalizable representations. In our experiments, ECG-JEPA$_{rb}$ and ECG-JEPA$_{mb}$ refer to ECG-JEPA models trained using random masking and multi-block masking strategies, respectively.

Table 1: Linear evaluation on multi-label and multi-class tasks. Our proposed method outperforms all baselines, achieving the highest AUC and F1 scores across both tasks and datasets.

| | | Multi-label Task | | | | Multi-class Task | | | |
| | | PTB-XL | | CPSC2018 | | PTB-XL | | CPSC2018 | |
| Method | Epochs | AUC | F1 | AUC | F1 | AUC | F1 | AUC | F1 |
|---|---|---|---|---|---|---|---|---|---|
| ST-MEM | 800 | 0.880 | 0.640 | 0.963 | 0.756 | 0.866 | 0.528 | 0.973 | 0.752 |
| SimCLR | 300 | 0.866 | 0.624 | 0.890 | 0.523 | 0.842 | 0.496 | 0.918 | 0.624 |
| CMSC | 300 | 0.802 | 0.472 | 0.767 | 0.206 | 0.796 | 0.442 | 0.787 | 0.391 |
| CPC | 100 | 0.620 | 0.167 | 0.687 | 0.091 | 0.600 | 0.201 | 0.672 | 0.210 |
| MoCo v3[1] | 800 | - | - | - | - | 0.739 | 0.142 | 0.712 | 0.080 |
| MTAE[1] | 800 | - | - | - | - | 0.807 | 0.437 | 0.818 | 0.349 |
| MLAE[1] | 800 | - | - | - | - | 0.779 | 0.382 | 0.794 | 0.263 |
| ECG-JEPA$_{rb}$ | 100 | 0.906 | 0.690 | 0.969 | 0.769 | 0.894 | 0.616 | **0.974** | 0.805 |
| ECG-JEPA$_{mb}$ | 100 | **0.912** | **0.712** | **0.971** | **0.789** | **0.896** | **0.628** | 0.973 | **0.819** |

[1] Scores reported in Na et al. (2024); results for multi-label tasks were not available.

Table 2: Reduced lead evaluation. Linear evaluation of PTB-XL multi-label classification in single-leade (II) and dual-lead (II and V1).

| | 1-Lead | | 2-Lead | |
| Method | AUC | F1 | AUC | F1 |
|---|---|---|---|---|
| ST-MEM | 0.832 | 0.571 | 0.840 | 0.573 |
| ECG-JEPA$_{rb}$ | 0.846 | **0.596** | 0.877 | 0.647 |
| ECG-JEPA$_{mb}$ | **0.849** | 0.593 | **0.880** | **0.657** |

## 5.1 Linear Evaluation

Table 1 present the results of our linear evaluation on the *PTB-XL* and *CPSC2018* datasets. We train a linear classifier on top of the frozen representations for 10 epochs and evaluate its performance on downstream tasks. Further training beyond 10 epochs does not lead to any significant improvement in performance. As shown in the tables, ECG-JEPA consistently outperforms other SSL methods, demonstrating superior efficiency and effectiveness with substantially reduced computational resources.

## 5.2 Reduced Lead Evaluation

To evaluate ECG-JEPA's performance under reduced input settings, we leveraged the flexibility of transformer architectures to handle variable input lengths. In this experiment, we conducted a linear evaluation on the *PTB-XL* multi-label task using only a single lead (Lead II) and two leads (Lead II and V1), training linear classifiers on the learned representations for 10 epochs[1]. Table 2 presents the results. Notably, ECG-JEPA maintains strong performance even with fewer leads, which is valuable for practical applications in mobile health monitoring, where most devices typically output only one or two leads.

## 5.3 Low-shot Linear Evaluation

Table 3 presents the performance comparison on the low-shot task. Low-shot learning is particularly challenging, as models must generalize effectively with limited labeled data. Given the difficulty and resource-intensive nature of obtaining labeled data in medical research, low-shot learning represents a realistic and critical scenario in the medical field. In this experiment, we evaluate the performance of ECG-SSL models on the *PTB-XL* multi-label task with only 1% and 10% of the training set, while keeping the test set fixed. As shown in the table, ECG-JEPA demonstrates a clear advantage over other SSL methods, with its effectiveness becoming particularly evident in low-shot learning tasks. This suggests that ECG-JEPA can be particularly well-suited for transfer learning where labeled data is scarce.

---

[1]We compare only with ST-MEM, as it is a transformer-based model whose pretrained weights are publicly available.

Table 3: Low-shot linear evaluation on the multi-label PTB-XL. The mean and standard deviation of macro AUCs are reported for 1% (192 samples) and 10% (1923 samples) of the training set, selected three times independently.

| Method | Epochs | PTB-XL 1% | PTB-XL 10% |
|---|---|---|---|
| ST-MEM | 800 | $0.807 \pm 0.003$ | $0.858 \pm 0.001$ |
| SimCLR | 300 | $0.803 \pm 0.002$ | $0.843 \pm 0.001$ |
| CMSC | 300 | $0.750 \pm 0.008$ | $0.792 \pm 0.001$ |
| CPC | 100 | $0.523 \pm 0.006$ | $0.560 \pm 0.005$ |
| ECG-JEPA$_{rb}$ | 100 | $\underline{0.836 \pm 0.006}$ | $\underline{0.887 \pm 0.000}$ |
| ECG-JEPA$_{mb}$ | 100 | $\mathbf{0.843 \pm 0.004}$ | $\mathbf{0.894 \pm 0.003}$ |

Table 4: ECG feature prediction results on PTB-XL multi-lable test set. The mean heart rate and QRS duration in the test set are 70.01 BPM ($\pm 17.65$) and 90.48 ms ($\pm 17.02$), respectively.

| Method | Mean Absolute Error Heart Rate (BPM) | Mean Absolute Error QRS Dur. (ms) |
|---|---|---|
| ST-MEM | $1.48 \pm 2.70$ | $4.94 \pm 4.54$ |
| SimCLR | $1.87 \pm 2.81$ | $6.14 \pm 5.80$ |
| CMSC | $7.20 \pm 7.43$ | $10.12 \pm 9.98$ |
| CPC | $11.40 \pm 11.04$ | $11.55 \pm 11.55$ |
| ECG-JEPA$_{rb}$ | $\underline{1.54 \pm 2.62}$ | $\underline{4.81 \pm 4.29}$ |
| ECG-JEPA$_{mb}$ | $\mathbf{1.45 \pm 2.44}$ | $\mathbf{4.41 \pm 4.08}$ |

### 5.4 ECG Feature Extraction

Extracting ECG features is crucial for diagnosing and monitoring cardiac conditions. In this experiment, we assess the model's ability to extract key features such as heart rate and average QRS duration from the learned representations. Unlike classification tasks, which focus on perceptual patterns, ECG features are directly tied to the signal's morphology.

Various methods exist for segmenting ECG signals (Sereda et al., 2019; Moskalenko et al., 2020; Chen et al., 2023; Joung et al., 2024), which can be used to extract ECG features. For this experiment, we utilized a publicly available segmentation model (Joung et al., 2024) to generate ground truth labels for heart rate and QRS duration from the PTB-XL dataset.

To compute the heart rate, the segmentation model identifies R peaks and calculates the average RR interval across the ECG signal. The heart rate is then derived using the formula $1000 \times (60/\text{avg RR interval})$, where the RR interval is expressed in milliseconds.

For an average QRS duration, the segmentation model detects the onset and offset of each QRS interval within the ECG. The duration of each QRS interval is computed as the difference between its offset and onset. The average QRS duration is then calculated as the mean of all detected QRS duration. We then trained a linear regression model on the learned representations to predict these features, using mean squared error (MSE) as the loss function.

Table 4 shows the performance comparison, reporting the means and standard deviations of the absolute differences between the predicted and extracted values for the heart rate and QRS duration across the PTB-XL test set.

### 5.5 Robustness Under Noise

The robustness of the proposed model was further evaluated across varying noisy conditions. Specifically, the model's performance was compared in three scenarios: (1) with basic preprocessing steps applied to remove noise (high-pass and low-pass filtering; noise level 0), (2) without preprocessing, retaining the inherent noise

present in raw signals (noise level 1), and (3) with artificially introduced noise (noise level 2). See Figure 12 for visualization of each noise level.

To simulate realistic noise, we incorporated two common ECG artifacts: baseline drift and powerline interference. These artifacts were generated using mathematical models to evaluate the model's performance under challenging conditions. Detailed explanations of the preprocessing steps and the artificial noise generation are provided in Appendix C.4.

Figure 7 presents the performance of the top four models on the PTB-XL multi-label task. Notably, both ECG-JEPA and SimCLR demonstrate considerable robustness even under severe noise conditions (noise level 2), whereas the performance of ST-MEM drops significantly. This disparity can possibly be attributed to the fundamental differences in their approaches: ECG-JEPA and SimCLR are latent representation prediction models. In contrast, ST-MEM focuses on reconstructing the raw signal itself, making it more susceptible to noise. However, further investigation is required to confirm this hypothesis and to better understand the underlying factors contributing to noise robustness.

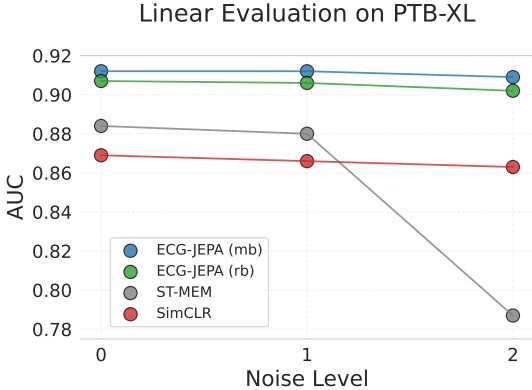

Figure 7: Performance comparison under varying noise levels on the PTB-XL multi-label task.

## 5.6 Fine-tuning

Fine-tuning is another method to evaluate the quality of learned representations, as it tests the model's ability to adapt its pre-trained features to new tasks. We add a linear classification head at the end of the encoder and train the entire network for 10 epochs. Similar to linear evaluation, training for 10 epochs is sufficient, as further training does not lead to additional performance gains. Fine-tuning can potentially enhance performance beyond what is achieved with linear evaluation alone.

To further boost performance during fine-tuning, preprocessing steps are applied on both training and test sets. Preprocessings include high-pass and low pass filterings, same preprocessings used in Section 5.5, which mitigate common ECG artifacts such as baseline drift and powerline interference noise.

Table 5 presents the results of fine-tuning on the *PTB-XL* and *CPSC2018* datasets. ECG-JEPA is compared with other SSL methods as well as supervised methods in a multi-class classification setting, where the student network is trained directly from the scratch. The results indicate that ECG-JEPA achieves the highest AUC and F1 scores on *PTB-XL* and the highest AUC on *CPSC2018*.

## 5.7 Effect of CroPA

Table 6 presents the results of our evaluation of the effectiveness of CroPA. CroPA introduces a "human-like" inductive bias, enabling the model to be trained more efficiently on multi-lead ECG data. Without CroPA, models may require more epochs to converge. For a fair comparison, we trained ECG-JEPA with and without CroPA for 100 and 200 epochs and compared their performance on the PTB-XL multi-class task. The results show that CroPA improves the model's performance, demonstrating its effectiveness in capturing inter-lead relationships and enhancing the model's ability to learn meaningful representations.

# 6 Visualization of ECG Representations

Dimensionality reduction techniques enable the visualization of high-dimensional datasets, providing valuable insights into uncovering hidden patterns within complex data. UMAP (McInnes et al., 2018), a widely used non-linear dimensionality reduction method, balances local versus global structure in the data.

In this section, we employ UMAP to visualize two prominent rhythm categories from PTB-XL: normal sinus rhythm (NSR) and atrial fibrillation (AFib). These labels comprise 16,687 samples (train: 15,021; test: 1,666) and 1,514 samples (train: 1,335; test: 149) in the rhythm category, respectively. See Appendix C.1 for further explanation on the dataset. SR is characterized by a regular rhythm and a single P wave for each QRS complex, whereas AFib is characterized by irregular and often rapid heart rhythms. Although AFib is not directly related to diagnostic statements of the heart, it significantly increases the risk of stroke, heart failure, and other cardiovascular complications.

Figure 8 illustrates the UMAP projection of the NSR and AFib samples from the test set, where UMAP is fitted on SR and AFib samples from the train set. The majority of NSR ECGs (orange) and AFib ECGs (blue) are well-separated in the 2D space, though a few samples overlap with different clusters. These patterns highlight the need for further exploratory data analysis to better understand the structure and quality of the dataset. Notably, overlapping samples or outliers in unexpected clusters may indicate mislabeled instances. Such cases are examined in detail in Appendix E to identify opportunities for enhancing the dataset's quality. This analysis demonstrates the potential of our model to aid in refining large-scale clinical datasets by uncovering hidden data issues.

# 7 Broader Impact and Ethical Considerations

This work focuses on developing a self-supervised learning framework for 12-lead ECGs using fully anonymized data, addressing privacy concerns. While our method is not a diagnostic tool, it enables more effective ECG representation learning for downstream tasks. Ensuring fairness, reliability, and clinical safety in applications built upon our model remains essential.

# 8 Conclusion

We proposed ECG-JEPA, an effective SSL method tailored for 12-lead ECG data. By utilizing latent space prediction architecture coupled with the innovative masked self-attention mechanism, CroPA, ECG-JEPA effectively learns meaningful representations of ECG signals. This approach addresses the challenges posed by noise and artifacts in ECG data, demonstrating substantial improvements over existing SSL methods in various downstream tasks, with the added benefit of significantly faster convergence.

Table 5: Fine-tuning on multi-class task.

| Method | Epochs | PTB-XL | | CPSC2018 | |
|---|---|---|---|---|---|
| | | AUC | F1 | AUC | F1 |
| Supervised | 100 | 0.887 | 0.608 | 0.893 | 0.566 |
| MoCo v3[1] | 800 | 0.913 | 0.644 | 0.967 | **0.838** |
| MTAE[1] | 800 | 0.910 | 0.613 | 0.961 | 0.769 |
| MLAE[1] | 800 | 0.915 | 0.625 | 0.973 | 0.816 |
| CMSC[1] | 800 | 0.877 | 0.510 | 0.938 | 0.717 |
| ST-MEM | 800 | 0.929 | 0.668 | 0.977 | 0.820 |
| SimCLR | 300 | 0.905 | 0.650 | 0.934 | 0.693 |
| CPC[2] | 100 | - | - | - | - |
| ECG-JEPA$_{rb}$ | 100 | **0.944** | **0.710** | 0.980 | 0.821 |
| ECG-JEPA$_{mb}$ | 100 | 0.937 | 0.680 | **0.983** | 0.799 |

[1] Scores reported in Na et al. (2024).
[2] We did not fine-tune CPC due to its slow training process.

Table 6: Effect of CroPA. Linear evaluation (*lin*) and fine-tuning (*ft*) results on PTB-XL multi-class task.

| Mask | CroPA | Epochs | *lin* AUC | *ft* AUC |
|------|-------|--------|-----------|----------|
| Random | x | 100 | 0.888 | 0.930 |
| Random | x | 200 | 0.887 | 0.927 |
| Random | o | 100 | **0.894** | **0.944** |
| Multi-block | x | 100 | 0.872 | 0.924 |
| Multi-block | x | 200 | 0.886 | 0.914 |
| Multi-block | o | 100 | **0.896** | **0.937** |

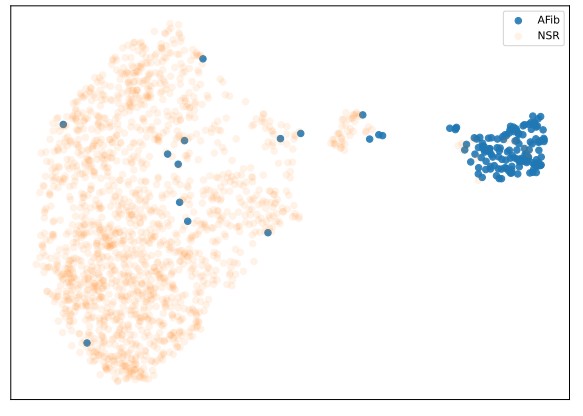

(a) NSR-focused Opacity        (b) AFib-focused opacity

Figure 8: UMAP visualization of ECG representations (NSR and AFib) from the PTB-XL test set.

Our extensive experimental evaluations reveal that ECG-JEPA outperforms state-of-the-art SSL methods across several tasks, including linear evaluation, fine-tuning, low-shot learning, and ECG feature extraction. Moreover, our investigation into the use of 8 leads, as opposed to the full 12-lead ECG, indicates that this reduction does not compromise performance while optimizing computational efficiency. This finding is particularly significant for applications constrained by limited computational resources.

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

# Appendix

## A  Additional Experiments

### A.1  Pretraining on a Larger ECG Dataset

Recent advances in machine learning have demonstrated that model performance often follows predictable scaling laws: as the size of the model and/or dataset increases, performance typically improves following a power-law relationship Kaplan et al. (2020). This observation, originally reported in the context of natural language processing and computer vision tasks, motivates the investigation of whether similar trends hold in the ECG domain.

In our work, we sought to explore the impact of incorporating a larger dataset into ECG pretraining. However, the availability of large, open ECG datasets remains limited. With the exception of the MIMIC-IV-ECG v1.0 dataset—which initially contains approximately 800,000 12-lead, 10-second ECGs—most publicly available ECG datasets are relatively small. After excluding roughly 20,000 ECGs with missing values, about 780,000 samples were used in pretraining. Although MIMIC-IV-ECG offers a substantial amount of data, many of these ECGs are obtained from hospital admissions, emergency departments, and intensive care units, implying a bias towards more acute or critical conditions.

To evaluate the effect of this larger dataset on pretraining performance, we incorporated the MIMIC-IV-ECG dataset into our pretraining pipeline alongside our original datasets (Chapman, Ningbo, and CODE-15), and then assessed the resulting model on downstream tasks using the PTB-XL and CPSC2018 datasets. In this preliminary study, we added the larger dataset all at once rather than incrementally, which may limit the precision of our analysis of scaling effects. Additionally, we did not increase the model size due to limited computational resources.

Tables 7 and 8 provide comparisons of ECG-JEPA's performance when pretrained on either the original datasets or an extended dataset that includes the extensive MIMIC-IV-ECG collection. Notably, linear evaluation results show a slight drop in performance with the extended pretraining dataset, whereas fine-tuning performance remains very similar between the two settings. Despite the shift in pretraining data distribution, our findings indicate that the inclusion of MIMIC-IV-ECG data does not significantly degrade downstream performance, suggesting that the model is capable of extracting robust features even when trained on large datasets with inherent biases. These preliminary results warrant further investigation to fully understand the underlying dynamics and to optimize pretraining strategies for ECG data.

Table 7: Linear evaluation of ECG-JEPA using original (180k) vs. extended (960k) pretraining datasets

| Method | Pretrain Size | Multi-label Task | | | | Multi-class Task | | | |
| | | PTB-XL | | CPSC2018 | | PTB-XL | | CPSC2018 | |
| | | AUC | F1 | AUC | F1 | AUC | F1 | AUC | F1 |
| ECG-JEPA$_{rb}$ (Original) | 180k | 0.906 | 0.690 | 0.969 | 0.769 | 0.894 | 0.616 | 0.974 | 0.805 |
| ECG-JEPA$_{rb}$ (Extended) | 960k | 0.898 | 0.679 | 0.972 | 0.775 | 0.881 | 0.593 | 0.968 | 0.756 |

Table 8: Fine-tuning performance of ECG-JEPA on multi-class classification using original (180k) vs. extended (960k) pretraining datasets.

| Method | Pretrain Size | PTB-XL | | CPSC2018 | |
| | | AUC | F1 | AUC | F1 |
| ECG-JEPA$_{rb}$ (Original) | 180k | 0.944 | 0.710 | 0.980 | 0.821 |
| ECG-JEPA$_{rb}$ (Extended) | 960k | 0.943 | 0.732 | 0.980 | 0.822 |

Table 9: Linear evaluation tasks using a dedicated validation set for hyperparameter selection.

| Method | Multi-label Task | | | | Multi-class Task | | | |
| | PTB-XL | | CPSC2018 | | PTB-XL | | CPSC2018 | |
| | AUC | F1 | AUC | F1 | AUC | F1 | AUC | F1 |
|---|---|---|---|---|---|---|---|---|
| ST-MEM | 0.881 | 0.625 | 0.960 | 0.695 | 0.874 | 0.577 | 0.963 | 0.788 |
| SimCLR | 0.871 | 0.597 | 0.918 | 0.568 | 0.835 | 0.504 | 0.921 | 0.586 |
| ECG-JEPA$_{rb}$ | 0.906 | 0.677 | 0.979 | 0.774 | 0.896 | 0.659 | **0.970** | 0.784 |
| ECG-JEPA$_{mb}$ | **0.913** | **0.707** | **0.976** | **0.807** | **0.904** | **0.639** | 0.979 | **0.833** |

Table 10: Fine-tuning for multi-class classification with hyperparameters selected using a dedicated validation set.

| Method | PTB-XL | | CPSC2018 | |
| | AUC | F1 | AUC | F1 |
|---|---|---|---|---|
| Supervised | 0.882 | 0.589 | 0.892 | 0.583 |
| ST-MEM | 0.910 | 0.617 | 0.977 | 0.820 |
| SimCLR | 0.928 | 0.677 | 0.955 | 0.682 |
| ECG-JEPA$_{rb}$ | **0.942** | **0.697** | 0.980 | 0.821 |
| ECG-JEPA$_{mb}$ | 0.933 | 0.663 | **0.983** | 0.799 |

## A.2 Validation-Based Performance Evaluation

Although each model in our experiments used the same hyperparameter settings, concerns may still arise regarding potential bias from not using a dedicated validation set for hyperparameter tuning. To address these concerns, we adopt a strict validation-based evaluation framework. This section explains our procedure for data splitting, hyperparameter tuning, and performance evaluation, which together help us reliably assess the generalization ability of our models.

For the PTB-XL dataset, we leverage the natural train, validation, and test splits provided by the dataset. In contrast, the CPSC2018 dataset does not come with predefined splits; hence, we randomly partition it into training, validation, and test sets using an 8:1:1 ratio.

For both datasets, we performed a grid search over 10 different learning rate values, logarithmically spaced between 0.1 and 0.0001. The best-performing learning rate was selected based on the validation set AUC score. Models were trained for 100 epochs, with early stopping applied (patience of 10 epochs) to prevent overfitting based on the validation AUC score. Additionally, we used the AdamW optimizer with its default weight decay of 0.01.

Tables 9 and 10 summarize the performance of our methods under this evaluation framework. Notably, using a dedicated validation set for hyperparameter tuning did not result in a performance drop, further confirming the robustness of our evaluation approach.

## A.3 Nearest Neighborhood Classifier

While linear probing and fine-tuning are common ways to evaluate SSL models, *Nearest Neighborhood Classifier (NCC)* is the simplest way to evaluate the SSL models without further training. NCC is a light-weight multi-class classifier that does not involve training any models, and the class of a test sample is determined by the distribution of the training samples. Specifically, let $\{(x_i, y_i)\}$ be the set of pairs of training sample and its label. Assuming that there are $C$ classes, we compute the class-mean vectors in the training set:

$$\mu_c = \frac{1}{|\{y_i = c\}|} \sum_{y_i = c} x_i, \quad c \in [C].$$

Table 11: NCC classifier on several datasets in multi-class task.

| Method | Distance | PTB-XL | | CPSC2018 | |
|---|---|---|---|---|---|
| | | Acc. | F1 | Acc. | F1 |
| ST-MEM | Euclidean | 0.524 | 0.419 | 0.611 | 0.571 |
| SinCLR | Euclidean | 0.567 | 0.452 | 0.498 | 0.443 |
| ECG-JEPA$_{rb}$ | Euclidean | **0.609** | **0.489** | 0.707 | 0.675 |
| ECG-JEPA$_{mb}$ | Euclidean | 0.584 | 0.446 | **0.676** | 0.644 |
| ST-MEM | Cosine | 0.524 | 0.420 | 0.613 | 0.574 |
| SinCLR | Cosine | 0.560 | 0.445 | 0.497 | 0.440 |
| ECG-JEPA$_{rb}$ | Cosine | **0.604** | **0.484** | **0.705** | **0.670** |
| ECG-JEPA$_{mb}$ | Cosine | 0.577 | 0.444 | 0.672 | 0.640 |

Table 12: Linear evaluation on G12EC multi-label task.

| Methods | AUC | F1 |
|---|---|---|
| ST-MEM | 0.894 | 0.406 |
| SimCLR | 0.859 | 0.276 |
| ECG-JEPA$_{rb}$ | 0.922 | 0.493 |
| ECG-JEPA$_{mb}$ | 0.927 | 0.507 |

The test sample $x$ is then classified according to the closest class-mean vectors, where the distance can be either Euclidean or cosine similarity. This method is similar to the $k$-nearest neighbor classifier, but it is simpler because it does not involve choice of $k$.

Table 11 shows the result of NCC results across multiple datasets. Note that we cannot compute the AUC as NCC does not produce class probabilities. While ECG-JEPA still outperforms other method, two points are worth mentioning. (1) Unlike linear evaluation, ECG-JEPA$_{rb}$ performs better than ECG-JEPA$_{mb}$, and more interestingly, (2) the performance of SimCLR drops drastically in CPSC2018 dataset. While further analysis is needed to confirm the reason, the authors suspect that the class-mean vectors for SimCLR representations are not robust because per-class samples are much smaller in size compared to PTB-XL dataset.

## A.4 Linear Evaluation on the Georgia ECG Dataset

The Georgia 12-lead ECG Challenge (G12EC) Database is part of the dataset used in the PhysioNet/Computing in Cardiology Challenge 2020 (Alday et al., 2020), a multi-label classification challenge with 27 labels. Although the G12EC dataset is designed with separate training, validation, and test sets, only the training set is publicly available; henceforth, we refer to it as the open G12EC training dataset. This dataset consists of 10,292 10-second 12-lead ECG recordings sampled at 500 Hz. We further partition the open G12EC training dataset into training, validation, and test sets using an 8:1:1 ratio.

Table 12 reports the linear evaluation performance of SSL models on top of fixed representations. As described in Appendix A.3, we train a linear classifier for 100 epochs with early stopping (patience of 10 epochs) based on the validation set AUC. Additionally, we perform a grid search over 10 different learning rate values, logarithmically spaced between 0.1 and 0.0001, using the validation set to select the best-performing learning rate.

## A.5 CroPA's Effect: Statistical Significance Analysis

To rigorously assess the impact of CroPA, we perform a statistical significance test using linear probes. We compare two pretrained models of ECG-JEPA$_{rb}$: one pretrained with CroPA and the other without CroPA. We bootstrap the differences in AUC,

$$\Delta \text{AUC} = \text{AUC}_{\text{CroPA}} - \text{AUC}_{\text{noCroPA}},$$

by randomly sampling (with replacement) from the test set of each dataset, using a sample size equal to that of the full test set. This process is repeated 200 times to compute the 95% confidence interval of $\Delta$AUC.

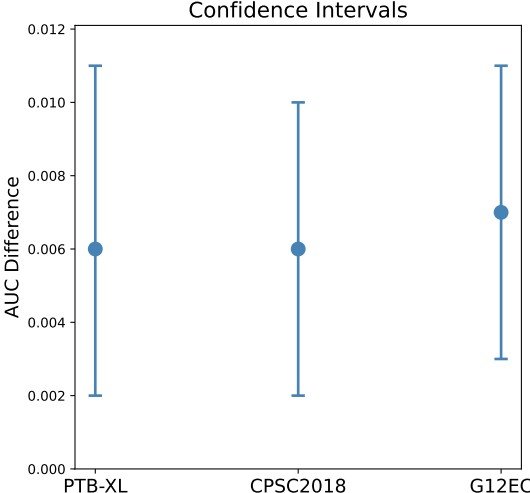

Figure 9: Bootstrapped 95% confidence intervals for the difference in AUC ($\Delta$AUC) between models pre-trained with and without CroPA on the PTB-XL, CPSC2018, and G12EC datasets. The strictly positive intervals, even though the observed improvements are small, demonstrate that CroPA yields a statistically significant improvement in performance.

As shown in Figure 9, the bootstrapped 95% confidence intervals for the AUC differences ($\Delta$AUC) between models pretrained with and without CroPA indicate a statistically significant improvement in performance. Although the improvement is modest, the strictly positive confidence intervals across all three datasets support the conclusion that CroPA yields a statistically significant improvement in performance.

# B Ablation Study

## B.1 Masking Ratio

Table 13 presents the performance of ECG-JEPA in linear evaluation with different masking ratios and strategies. The results indicate that the model benefits from a high masking ratio. Notably, multi-block masking is advantageous for linear evaluation, while random masking is more effective for fine-tuning, as indicated in Table 5. Although random masking with a ratio of (0.7, 0.8) achieves better performance in the PTB-XL multi-label task, a masking ratio of (0.6, 0.7) performs better in other tasks. Therefore, we chose the latter for our main experiments.

Table 13: Effect of masking strategy. Linear evaluation results on PTB-XL multi-label task using different masking ratios and strategies.

| Mask | Ratio | Freq. | AUC | F1 |
|---|---|---|---|---|
| Random | (0.3, 0.4) | 1 | 0.884 | 0.652 |
| Random | (0.4, 0.5) | 1 | 0.904 | 0.698 |
| Random | (0.5, 0.6) | 1 | 0.906 | 0.697 |
| Random | (0.6, 0.7) | 1 | 0.906 | 0.690 |
| Random | (0.7, 0.8) | 1 | **0.909** | **0.706** |
| Multi-block | (0.10, 0.15) | 4 | 0.904 | 0.678 |
| Multi-block | (0.15, 0.20) | 4 | 0.905 | 0.687 |
| Multi-block | (0.175, 0.225) | 4 | **0.912** | **0.712** |

## B.2 Comparison with 12-Lead Model

We now investigate the practical sufficiency of using 8 leads for ECG-JEPA pretraining. To evaluate the impact of this reduction, we trained models using both 8 leads and 12 leads and compared their performance on the linear evaluation of a multi-label task for PTB-XL.

Table 14 presents the results of this comparison using ECG-JEPA$_{rb}$. As expected, the performance difference between the 8-lead and 12-lead models is minimal, indicating that using 8 leads is sufficient for effective pretraining without significant loss of information.

Table 14: Comparison of 8-Lead and 12-Lead Models on PTB-XL multi-label.

| Model | epochs | AUC | F1 |
|---|---|---|---|
| 8-Lead | 100 | 0.906 | 0.690 |
| 12-Lead | 100 | 0.905 | 0.699 |

# C Experimental Details

## C.1 Downstream Datasets Details

Table 15: PTB-XL Diagnostic Category Distribution.

| Type | Set | # ECG | Norm | MI | STTC | CD | HYP |
|---|---|---|---|---|---|---|---|
| Multi-label | Total | 21799 | 9514 | 5469 | 5235 | 4898 | 2649 |
| | Train | 19230 | 8551 | 4919 | 4714 | 4402 | 2387 |
| | Test | 2158 | 963 | 550 | 521 | 496 | 262 |
| Multi-class | Total | 16244 | 9069 | 2532 | 2400 | 1708 | 535 |
| | Train | 14594 | 8157 | 2276 | 2158 | 1524 | 479 |
| | Test | 1650 | 912 | 256 | 242 | 184 | 56 |

Table 16: PTB-XL Rhythm Category Distribution.

| Type | Set | # ECG | NSR | AFib | Others |
|---|---|---|---|---|---|
| Multi-label | Total | 21030 | 16748 | 1514 | 2912 |
| | Train | 18932 | 15074 | 1362 | 2625 |
| | Test | 2098 | 1674 | 152 | 287 |
| Multi-class | Total | 20887 | 16687 | 1484 | 2716 |
| | Train | 18804 | 15021 | 1335 | 2448 |
| | Test | 2083 | 1666 | 149 | 268 |

Tables 15, 16 and 17 show the distribution of the PTB-XL diagnostic, rhythm, and CPSC2018 datasets, respectively. Note that the sum of samples in each class exceeds the total number of ECG recordings in multi-label task.

The PTB-XL dataset is stratified into ten folds, where the first eight folds are used for training, the ninth fold for validation, and the tenth fold for testing. In our experiments, we used the first nine folds for training and the tenth fold for testing, as we did not observe overfitting during linear evaluation and fine-tuning.

For the CPSC2018 dataset, only the training set is publicly available, which is stratified into seven folds. We used the first six folds for training and the seventh fold for testing, omitting the validation set. The original CPSC2018 dataset consists of 6,877 ECG recordings, but we excluded recordings with a length of less than 10 seconds, resulting in 6,867 ECG recordings.

Table 17: CPSC2018 Distribution.

| Type | Set | # ECG | Norm | PVC | AF | LBBB | STE | 1AVB | PAC | STD | RBBB |
|------|-----|-------|------|-----|----|----|-----|------|-----|-----|------|
| Multi-label | Total | 6867 | 918 | 1220 | 235 | 220 | 721 | 614 | 699 | 868 | 1854 |
| | Train | 5989 | 805 | 1059 | 206 | 197 | 632 | 534 | 615 | 742 | 1616 |
| | Test | 878 | 113 | 161 | 29 | 23 | 89 | 80 | 84 | 126 | 238 |
| Multi-class | Total | 6391 | 918 | 975 | 178 | 185 | 685 | 531 | 606 | 783 | 1530 |
| | Train | 5577 | 805 | 849 | 159 | 169 | 600 | 459 | 534 | 671 | 1331 |
| | Test | 814 | 113 | 126 | 19 | 16 | 85 | 72 | 72 | 112 | 199 |

## C.2 Hyperparameters for ECG-JEPA

Hyperparameters for ECG-JEPA pretraining, linear evaluation, and fine-tuning are provided in Tables 18, 10, and 11, respectively. In ECG-JEPA$_{mb}$, the number of visible patches in ECG-JEPA$_{mb}$ varies more than in ECG-JEPA$_{rb}$, resulting in higher GPU memory usage. Consequently, we reduced the batch size to 64 to fit the model on a single NVIDIA RTX 3090 GPU. Interestingly, ECG-JEPA$_{mb}$ benefits from larger learning rates, even with the halved batch size.

For fine-tuning process, the actual learning rate is calculated as $lr = base\_lr \times batchsize/256$, following the heuristic by Goyal et al. (2018).

Table 18: Pretraining Settings for ECG-JEPA.

| config | ECG-JEPA$_{rb}$ | ECG-JEPA$_{mb}$ |
|--------|-----------------|-----------------|
| optimizer | AdamW | AdamW |
| learning rate | 2.5e-5 | 5e-5 |
| weight decay | 0.05 | 0.05 |
| batch size | 128 | 64 |
| learning rate schedule | cosine decay | cosine decay |
| warmup epochs | 5 | 5 |
| epochs | 100 | 100 |
| drop path | 0.1 | 0.1 |

Figure 10: Linear Evaluation Settings

| config | value |
|--------|-------|
| optimizer | AdamW |
| learning rate | 0.01 |
| weight decay | 0.05 |
| batch size | 32 |
| learning rate schedule | cosine decay |
| warmup epochs | 3 |
| epochs | 10 |

Figure 11: Fine-tuning Settings

| config | value |
|--------|-------|
| optimizer | AdamW |
| base learning rate | 1.0e-4 |
| weight decay | 0.05 |
| batch size | 16 |
| learning rate schedule | cosine decay |
| warmup epochs | 3 |
| epochs | 10 |

## C.3 Hyperparameters for Other Pretrained Models

Besides pretraining ECG-JEPA, we also pretrained other models, including CMSC (Kiyasseh et al., 2021), CPC (van den Oord et al., 2019), and SimCLR (Chen et al., 2020) using the same datasets as ECG-JEPA.

For CMSC and CPC, we adhered to the original architecture and hyperparameters. SimCLR utilized a ResNet50 (He et al., 2016) encoder with an output dimension of 2048. CMSC and SimCLR were pretrained for 300 epochs, selecting the best checkpoint at 100, 200, or 300 epochs based on linear evaluation performance on the PTB-XL multi-label setting. Due to the slow training process, CPC was pretrained for only 100 epochs, taking approximately 9 days on a single NVIDIA RTX 3090 GPU due to the LSTM module in the model. For ST-MEM, we employed the publicly available checkpoint pretrained for 800 epochs.

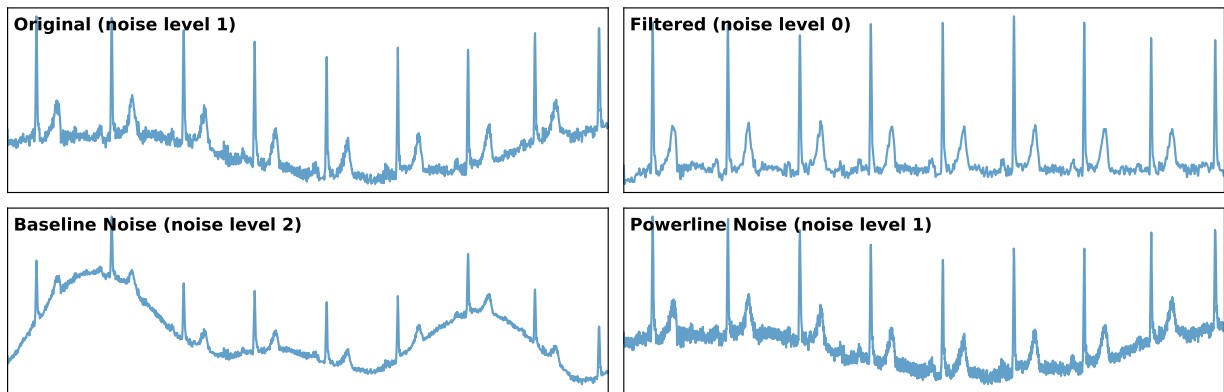

Figure 12: Visualization of ECG signals under the effect of filtering and added noise conditions. Note that the original signal contains both mild baseline and powerline noise.

Given SimCLR's sensitivity to data augmentations, we applied several that work well empirically: baseline shift (adding a constant to all leads), baseline wander (low-frequency noise), Gaussian noise (random noise), powerline noise (50 Hz noise), channel resize, random crop, and jump noise (sudden jumps). These augmentations aimed to enhance the robustness of the model to various signal distortions.

### C.4 Noise Generation and Preprocessing for ECG Signals

To evaluate the pretrained models' robustness under noise (Section 5.5), we preprocess ECGs to generate noise-removed data, and we add artificial noise to ECGs to generate ECGs with strong noise. Specifically, we apply high-pass and low-pass filters with cutoff frequencies 0.67 Hz and 40 Hz, respectively. This effectively removes both baseline drift and powerline interference noise.

While applying filters for noise removal is both straightforward and effective, generating realistic noise is more complex. Following Lenis et al. (2017), we use the following mathematical model to generate realistic baseline drift:

$$b(t) = C \cdot \sum_{k=0}^{K} a_k \cdot \cos\left(2\pi \cdot k \cdot \Delta f \cdot t + \phi_k\right)$$

with $\Delta f = f_s/N = 0.1\,\text{Hz}$, where $f_s = 250\,\text{Hz}$ is the sampling frequency and $N = 2500$ is the total number of time steps. Additionally, $K = 5$ represents the number of sinusoidal components, the amplitude coefficient $a_k$ is randomly sampled from a uniform distribution $[0, 1]$, while the phase $\phi_k$ is randomly drawn from the interval $[0, 2\pi)$. We use the scaling factor $C = 0.5$.

For powerline interference, we follow a noise generation approach inspired by Friesen et al. (1990). The powerline noise is modeled as a sum of sinusoidal components, including a base frequency $f_n = 50\,\text{Hz}$ and its higher harmonics. Specifically, given a sampling frequency $f_s = 250\,\text{Hz}$ and signal length $N = 2500$ timesteps, the noise is computed as:

$$s(t) = C \cdot \sum_{k=1}^{K} a_k \cdot \cos\left(2\pi k f_n t + \phi\right)$$

where $C = 0.5$ is a scaling factor, $a_k$ are random amplitude coefficients uniformly sampled from $[0, 1]$, and $\phi$ is randomly drawn from $[0, 2\pi)$. $K = 3$ specifies the number of higher harmonics considered.

Both types of noise were applied to all samples in the training and test sets with a probability of 0.5, and identical noise was added across all 8 leads. See Figure 12 for illustrations of the impact of high-pass and low-pass filtering, as well as the effect of added noise.

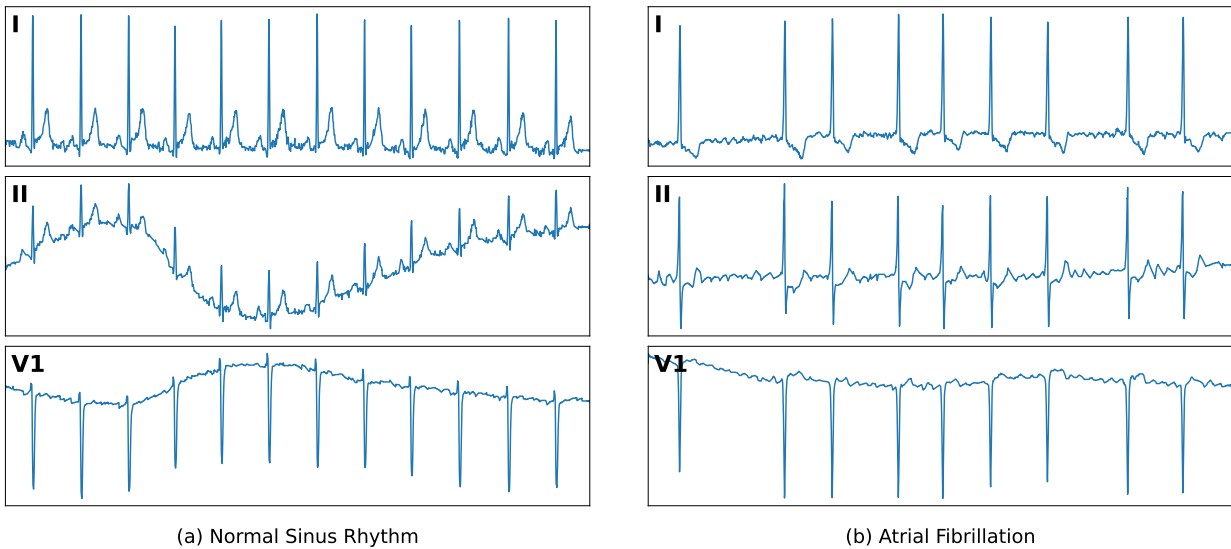

(a) Normal Sinus Rhythm            (b) Atrial Fibrillation

Figure 13: Comparison of NSR and AFib signals in leads I, II, and V1. **(a)** NSR demonstrates a regular heart rhythm with clear P waves. **(b)** AFib exhibits an irregular heart rhythm with the absence of P waves.

### C.5 Software Used in the Experiments

All experiments were conducted using Python 3.10 on an Ubuntu 20.04 operating system. The primary framework utilized was PyTorch 2.3 for model implementation and training, with CUDA 11.8 for GPU acceleration.

## D Exponential Moving Average

The teacher network is initialized as a copy of the student network and is updated using an exponential moving average (EMA) of the student's weights. The EMA is computed as follows:

$$\theta_{\text{teacher}}^i = \beta_i \theta_{\text{teacher}}^{i-1} + (1 - \beta_i)\theta_{\text{student}}^i$$

where $i$ denotes the current training iteration, and $\beta_i$ is a momentum parameter that evolves during training. The momentum parameter $\beta_i$ is computed as:

$$\beta_i = \text{ema}_0 + \frac{i \cdot (\text{ema}_1 - \text{ema}_0)}{\text{iterations\_per\_epoch} \cdot \text{epochs}}$$

Here, $\text{ema}_0$ and $\text{ema}_1$ represent the initial and final values of the momentum parameter, respectively. For our implementation, $\text{ema}_0 = 0.996$ and $\text{ema}_1 = 1.0$.

## E Case Analysis of UMAP Embeddings

In this section, we analyze individual ECG samples that are embedded in clusters different from their expected categories in the UMAP visualizations presented in Section 6. These cases include normal sinus rhythm (NSR) samples located within atrial fibrillation (AFib) clusters and AFib samples found in NSR clusters. Such occurrences provide valuable insights into the model's learned representations and highlight the challenges posed by atypical or borderline samples.

NSR typically exhibits a regular heart rhythm with distinct P waves preceding each QRS complex. In contrast, AFib is characterized by an irregular rhythm, the absence of discernible P waves, and the presence of fibrillatory waves—irregular, rapid oscillations of the baseline. Figure 13 consists of (a) an example of NSR

and (b) an example of AFib, illustrating the characteristic differences between the two. However, certain samples in the UMAP embeddings deviate from these standard definitions. To further understand these cases, we review the ECG signals of selected samples from each scenario.

## E.1 NSR Samples in AFib Clusters

Figure 14 shows an example of an NSR signal that is embedded in the AFib cluster. Upon inspection, this signal reveals irregularities in rhythm, and P waves are missing in leads V2-V6. These features, while atypical for NSR, may explain why the model's representation aligns this signal with the AFib cluster.

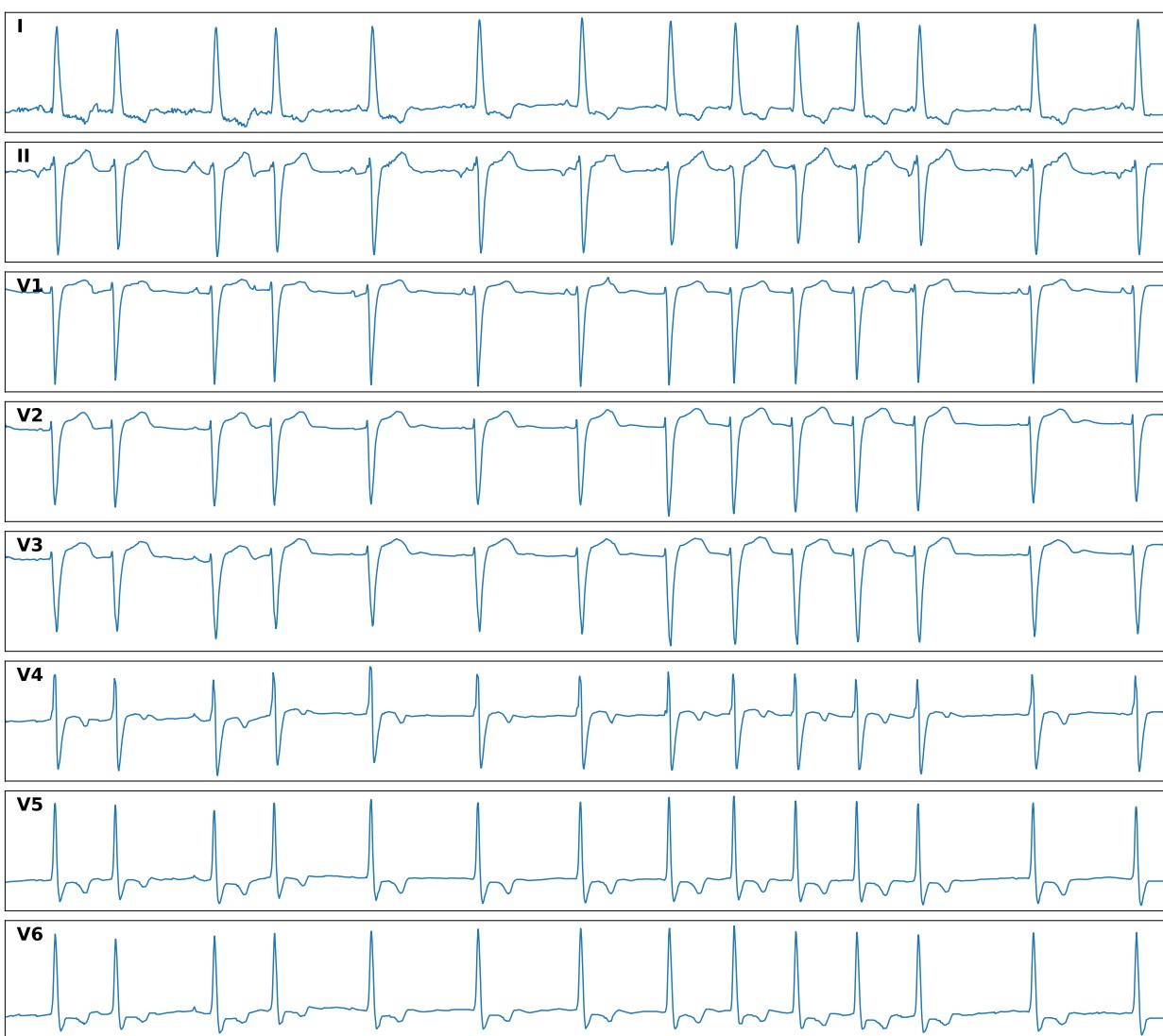

Figure 14: Example of an NSR signal embedded in the AFib cluster. The signal exhibits irregular rhythm and missing P waves in leads V2-V6, deviating from typical NSR characteristics.

### E.2 AFib Samples in NSR Clusters

Conversely, Figure 15 illustrates an AFib signal that is embedded in the NSR cluster. While this signal shows fibrillatory waves in leads I,II, and V1, the rhythm is regular and P waves are visible. This partial resemblance to NSR may have caused the model to assign it to the NSR cluster.

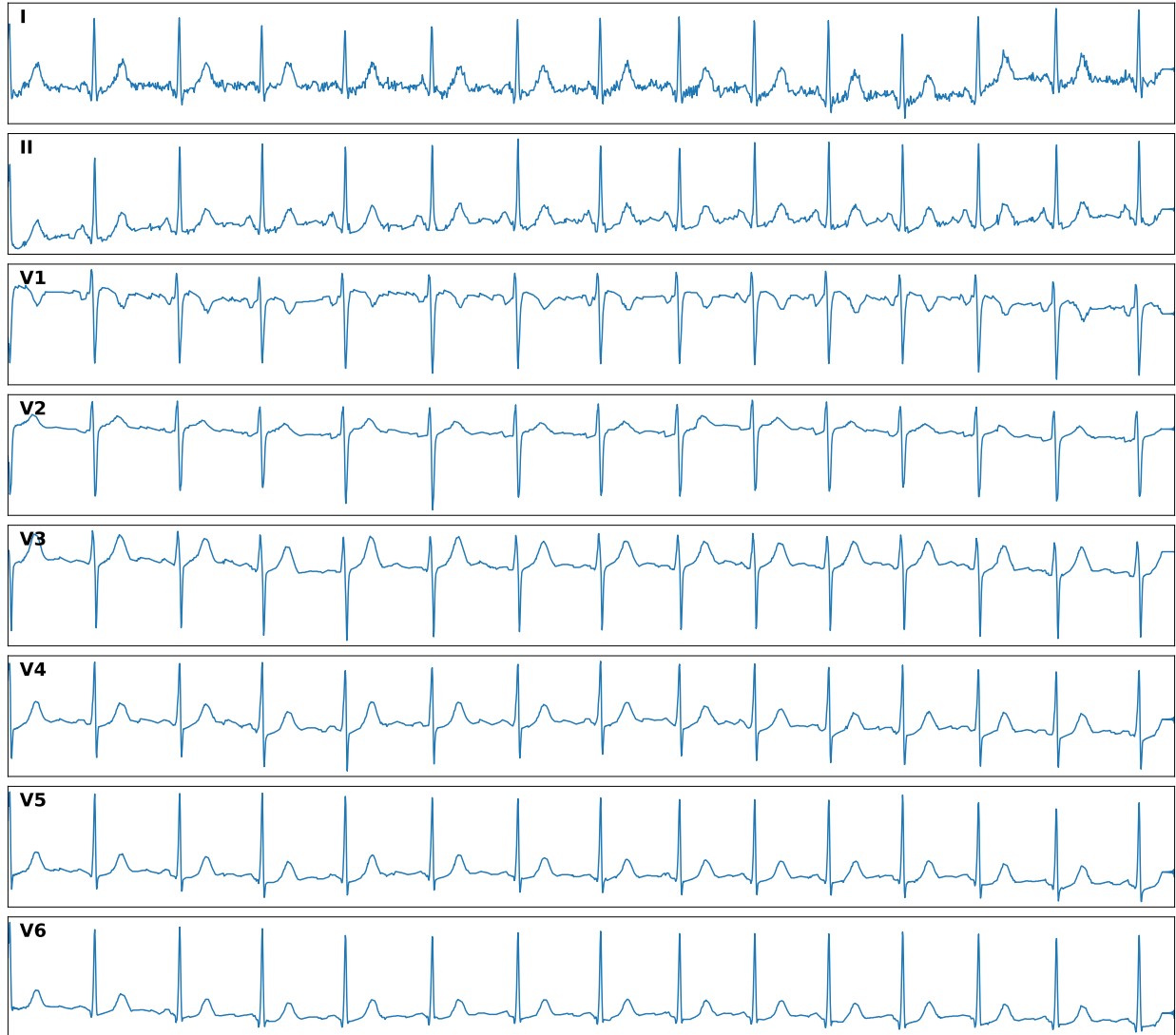

Figure 15: Example of an AFib signal embedded in the NSR cluster. The signal shows irregular P waves but exhibits a rhythm that mimics NSR to some extent.

### E.3 Implications of Atypical Cases

The presence of these atypical cases underscores the complexity of real-world ECG classification. Such samples may reflect physiological conditions that do not strictly align with the standard definitions of NSR or AFib, highlighting the potential for borderline or transitional states. Additionally, these cases might indicate mislabeled data, which is not uncommon given the inherent complexity of ECG interpretation.

Our analysis demonstrates that the model's learned representations are valuable not only for classifying typical cases but also for identifying and interpreting atypical cases. By examining UMAP embeddings, the model provides insights into ambiguous samples and helps uncover potential labeling inconsistencies. This

capability is particularly useful, as it can contribute to improving dataset quality by detecting and addressing mislabeled or borderline cases.

