# OpenReview forum: "Learning General Representation of 12-Lead Electrocardiogram With a Joint-Embedding Predictive Architecture"
_TMLR — Rejected by TMLR_

### Review · Reviewer_oSCd · 2025-01-24

**Summary Of Contributions:**

The authors introduced a JEPA SSL learning architecture designed for learning 12-lead ECG signal representations. They also proposed a CroPA-based attention mechanism tailored for multi-lead ECG data. The proposed algorithm demonstrated strong performance across several downstream tasks.

**Audience:**

Yes

**Broader Impact Concerns:**

N.A.

**Claims And Evidence:**

Yes

**Requested Changes:**

P2: "This is the first work to demonstrate that learned representations can effectively recover ECG features (Section 5.4)." This statement is vague and may be overclaimed. I suggest removing it. The training cost for ECG models is usually quite small due to the advancements in modern computing hardware. I recommend the authors place less emphasis on training cost.
Section 2.3: "reliance on LSTM modules makes it inefficient for large datasets." This statement is vague. Again, I suggest placing less emphasis on training cost.
Section 3.4: "This demands attention mechanisms that prioritize relationships within the same lead and within relevant time windows." This part lacks supporting evidence.
Section 3.4: "Specifically, a token xc,i attends xc′,i′ if and only if c = c′ (same lead) or i = i′ (same temporal space)." This explanation is unclear and needs clarification.
Section 6: "rhythm category" is not clearly defined.

Personal opinion (optional to consider): ECG signals are very different from voice and natural language signals due to their periodic nature and high temporal correlation. Given these characteristics, SSL might not be the most suitable approach for this domain.

**Strengths And Weaknesses:**

Strengths:
The results presented demonstrate a significant advantage over prior SSL-based approaches. The proposed model architecture is well-aligned with the unique properties of ECG data, showcasing its suitability for the task.

Weaknesses:
The clarity of the experimental methods in the main text could be further improved. While most details are available in the appendix, it would be beneficial to provide a clearer explanation in the main text. For example, in Section 5.6, it is unclear which data the "preprocessing steps" were applied to.
Additionally, I suggest including results from supervised training-based models to provide readers with better context and a more comprehensive comparison.

---

> ### Author Response · Authors · 2025-02-28
>
> **Comparison with Supervised Learning**
> > Additionally, I suggest including results from supervised training-based models to provide readers with better context and a more comprehensive comparison.
>
> We did not directly compare linear probing results with fully supervised training because, in linear evaluation, we only train a single linear classifier on top of the learned representations while keeping the encoder weights fixed. In contrast, supervised learning updates the entire model. Thus, it is not fair to compare linear evaluation results with fine-tuning outcomes.
>
> That being said, **we do compare with supervised learning in our fine-tuning experiments (Section 5.6)**, where the entire model is updated.
>
> ***
> **Training cost**
>
> We appreciate the reviewer’s suggestions regarding the emphasis on training cost.
>
> >The training cost for ECG models is usually quite small due to the advancements in modern computing hardware. I recommend the authors place less emphasis on training cost.
>
> While we agree that training costs have become significantly lower due to modern hardware and improved techniques, our focus on reducing training cost is somewhat orthogonal to these trends because our model benefits from fast convergence (only 100 epochs).
>
> > "reliance on LSTM modules makes it inefficient for large datasets." This statement is vague. Again, I suggest placing less emphasis on training cost.
>
> We acknowledge that the original statement was somewhat vague, and we have removed it in the revised version.
>
>
> ***
> **Clarity**
> > The clarity of the experimental methods in the main text could be further improved. While most details are available in the appendix, it would be beneficial to provide a clearer explanation in the main text. For example, in Section 5.6, it is unclear which data the "preprocessing steps" were applied to.
>
> We have clarified our experimental settings in the revised version. For instance, in Section 5.6, we now specify to which data the "preprocessing steps" were applied, as shown below:
>
> <em> "To further boost performance during fine-tuning, preprocessing steps are applied on both training and test sets. Preprocessings include high-pass and low pass filterings, same preprocessings used in Section 5.5, which mitigate common ECG artifacts such as baseline drift and powerline interference noise."</em>
>
> ***
> > This is the first work to demonstrate that learned representations can effectively recover ECG features (Section 5.4)." This statement is vague and may be overclaimed. I suggest removing it.
>
> We have removed that statement in the revised version.
>
> > Section 3.4: "This demands attention mechanisms that prioritize relationships within the same lead and within relevant time windows." This part lacks supporting evidence.
>
> > Section 3.4: "Specifically, a token xc,i attends xc′,i′ if and only if c = c′ (same lead) or i = i′ (same temporal space)." This explanation is unclear and needs clarification.
>
> We have revised Section 3.4 for improved clarity.
>
> > Section 6: "rhythm category" is not clearly defined.
>
> We appreciate you pointing out this important issue. We have revised Section 4.2, where we explain the downstream task datasets more clearly.
>
>
> ***
> **Others**
> > Personal opinion (optional to consider): ECG signals are very different from voice and natural language signals due to their periodic nature and high temporal correlation. Given these characteristics, SSL might not be the most suitable approach for this domain.
>
> Thank you for sharing your perspective. While we acknowledge that ECG signals differ from voice and natural language signals due to their periodic nature and high temporal correlation, we believe that self-supervised learning (SSL) remains highly relevant for ECG analysis for several reasons:
>
>
> 1. Generalization across datasets:  In a supervised setting, even when a model is trained on multiple datasets, it can still struggle to generalize to unseen datasets due to variations in data distribution, recording conditions, and patient populations. Rule-based classification is also challenging due to the diversity and complexity of ECG waveforms. SSL can help learn robust and transferable representations that mitigate these issues.
>
> 2. Beyond classification tasks: Our goal is to learn a general representation of ECG signals that is not limited to classification. Such representations can be beneficial in various downstream tasks, including multi-modal integration with other medical data (e.g., chest X-rays), thereby enabling richer and more contextualized clinical insights.

---

### Review · Reviewer_Co27 · 2025-02-08

**Summary Of Contributions:**

The work proposes an SSL framework, ECG-JEPA, for learning embeddings from ECG signals. ECG-JEPA consists of a teacher network, a student network and a predictor network. The student network takes a masked version of the ECG signals and feeds its output to the predictor network. The teacher network takes the clean version. The student and predictor networks are trained to reconstruct the latent representations produced by the teacher network. The work also proposes an attention mechanism called Cross-Pattern Attention which attends to signals from the same channel and other channel at the same step. The embeddings learned through ECG-JEPA shows strong performance against different baselines and robustness against noise.

**Audience:**

Yes

**Broader Impact Concerns:**

The work is an application-oriented work in an area with critical concerns on privacy and effectiveness. It should not be exempt from discussion on broader impact which is not present in the current work.

**Claims And Evidence:**

Yes

**Requested Changes:**

In addition to the broader impact concerns (see the section below), I suggest the following optional changes to make the work more solid:

1. The work compares against different baseline approaches but comparison against the same model learned directly through supervised learning on the downstream task is not made. Such comparison could better highlight the impacts of SSL pre-text training.

2. SSL used additional unlabeled data. The size of the pre-training data could have important impacts on model performance. I would strongly recommend the work to include experiments studying the effects of pre-training data size.

**Strengths And Weaknesses:**

**Strengths:**

As an application-oriented work, the work shows very solid and comprehensive experiment results in both quantitative studies and qualitative analysis and the studies shows strong performance against baseline methods.

**Weakness:**

The approach proposed my the work is essentially a masked auto-encoder, which has limited technical novelty. Claiming that CroPA is a inductive bias motivated by the characteristics of ECG data is also overreaching as similar attention mechanisms across different dimensions in time-series has been also studied in existing works [1, 2, 3].

[1] Kong, Zhifeng, et al. "Diffwave: A versatile diffusion model for audio synthesis." arXiv preprint arXiv:2009.09761 (2020).

[2] Tashiro, Yusuke, et al. "Csdi: Conditional score-based diffusion models for probabilistic time series imputation." Advances in Neural Information Processing Systems 34 (2021): 24804-24816.

[3] Shirzad, Hamed, et al. "Conditional diffusion models as self-supervised learning backbone for irregular time series." ICLR 2024 Workshop on Learning from Time Series For Health. 2024.

---

> ### Author Response · Authors · 2025-02-28
>
> We sincerely appreciate the reviewer’s thoughtful feedback and the opportunity to clarify the distinctions of our approach.
>
> ***
> **Comparison with masked autoencoder**
> > The approach proposed my the work is essentially a masked auto-encoder, which has limited technical novelty.
>
> While our method shares the general concept of masked modeling, labeling our approach as a masked autoencoder (MAE) overlooks key architectural and objective differences.
>
> 1. **Architectural Differences**
> MAEs rely on an encoder-decoder structure, where the encoder processes visible (unmasked) tokens and the decoder reconstructs the missing tokens in the original input space. In contrast, our model employs a student-teacher-predictor framework.
>
> 2. **Objective: generative v.s. non-generative**
> More importantly, a crucial difference is that our model does not aim to reconstruct the raw ECG signals. That is, MAE is considered to be generative as it reconstructs the signal. Instead, we predict the hidden representation of masked tokens rather than directly minimizing L1/L2 loss on the original signal. This decision is motivated by the inherent challenges in ECG reconstruction:
>
> 	- Small but clinically significant features, such as P-waves, can be overlooked by (L1/L2) losses in the raw data space. Since P-waves are small in amplitude, standard reconstruction losses may not fully differentiate between whether a P-wave exists or not.
>
> 	- Predicting in the raw signal space may result in unstable model behavior under noise that changes the signal shape significantly, as evidenced in Section 5.5.
>
> ***
> **CroPA**
>
> > Claiming that CroPA is an inductive bias motivated by the characteristics of ECG data is overreaching as similar attention mechanisms across different dimensions in time-series have been studied in existing works [1, 2, 3].
>
> While CroPA also leverages structured attention for multivariate time series, it differs from the mentioned papers [2, 3] in two key ways:
>
> 1. **Domain-Specific Inductive Bias**: CroPA is specifically designed for multi-lead ECG analysis, incorporating an inductive bias that aligns with how clinicians interpret ECG signals. This clinical practice can be found in [4], or in the answer to "Concern 2" from the reviewer fy8c.
> 2. **2D Attention vs. Consecutive 1D Attention**: Unlike in [2, 3] where consecutive 1D Transformer layers (first along time, then across channels) are applied sequentially, CroPA performs a single-step 2D attention operation that jointly captures both temporal and lead-wise dependencies. This allows CroPA to more directly enforce the clinical practice.
>
> We acknowledge that cross-dimensional attention (as in [2, 3]) has been widely explored in time-series modeling, and we appreciate the reviewer’s insights. Our intent with CroPA was to incorporate domain knowledge into the attention mechanism in a way that aligns with how ECG signals are typically analyzed. Additionally, we did not find any similar attention mechanism in [1].
>
> [1] Kong, Zhifeng, et al. "Diffwave: A versatile diffusion model for audio synthesis." arXiv preprint arXiv:2009.09761 (2020).
>
> [2] Tashiro, Yusuke, et al. "Csdi: Conditional score-based diffusion models for probabilistic time series imputation." Advances in Neural Information Processing Systems 34 (2021): 24804-24816.
>
> [3] Shirzad, Hamed, et al. "Conditional diffusion models as self-supervised learning backbone for irregular time series." ICLR 2024 Workshop on Learning from Time Series For Health. 2024.
>
> [4] Thaler, Malcolm S.  "The only EKG book you’ll ever need." Lippincott Williams & Wilkins, 2021. APA

---

> ### Author Response · Authors · 2025-02-28
>
> **Comparison with supervised learning**
>
> > The work compares against different baseline approaches, but a comparison against the same model learned directly through supervised learning on the downstream task is not made. Such a comparison could better highlight the impacts of SSL pretext training.
>
> We did not directly compare linear probing results with fully supervised training because, in linear evaluation, we only train a single linear classifier on top of the learned representations while keeping the encoder weights fixed. In contrast, supervised learning updates the entire model. Thus, it is not fair to compare linear evaluation results with fine-tuning outcomes.
>
> We did not directly compare linear probing results with fully supervised training because, in linear evaluation, we only train a single linear classifier on top of the learned representations while keeping the encoder weights fixed. In contrast, supervised learning updates the entire model. Thus, it is not fair to compare the results of linear evaluation with supervised training result.
>
> That being said, **we do compare with supervised learning in our fine-tuning experiments (Section 5.6)**, where the entire model is updated.
>
> ***
> **Pretrain on additional unlabeled data**
> > SSL used additional unlabeled data. The size of the pre-training data could have important impacts on model performance. I would strongly recommend the work to include experiments studying the effects of pre-training data size.
>
> Thank you for your suggestion. In response, we incorporated the MIMIC-IV-ECG v1.0 dataset—comprising roughly 800,000 12-lead, 10-second ECGs, predominantly from hospital inpatient, emergency, and intensive care unit settings—into our pretraining pipeline, increasing the dataset from 180k to 960k recordings.
>
> Our additional experiments (Tables 7, 8) indicate that while there is a slight drop in linear evaluation performance, fine-tuning results remain very similar. These findings suggest that despite the inherent bias of the larger dataset, our model is able to extract robust features. We think that further investigation is warranted to fully understand the scaling effects.
>
> ***
> **Broader Impact Concerns**
> > The work is an application-oriented work in an area with critical concerns on privacy and effectiveness. It should not be exempt from discussion on broader impact which is not present in the current work.
>
> We have added a new section (Section 7) in the revised paper that discusses broader impact concerns.

---

### Review · Reviewer_fy8c · 2025-02-19

**Summary Of Contributions:**

The paper proposes a self-supervised approach to learn representations of multivariate electrocardiogram (ECG) signals.

The primary claims made in the abstract/intro are that

1) The ECG-specific JEPA architecture leads to improved improved classification performance over other ECG SSL approaches
2) The method's novel use of cross-pattern attention (CroPA) is particularly key to improving downstream task performance
3) The method can be used to predict "ECG features" like heart rate or QRS duration, which hasn't been done by previous methods.
4) The method is "highly scalable", reaching similar or better AUC with fewer GPU hours of training time (Fig 3)

JEPA (joint embedding predictive architecture) is a general approach to self-supervised learning for arbitrary data types originally pioneered by Lecun (2022). This work has operationalized JEPA ideas for ECG data. The modeling strategy or "method" is illustrated in Figure 4. Short fragments (like 0.2 seconds) of a single ECG lead (aka channel) are treated as "patches". Each patch is mapped to a token vector in D dimensions by a "teacher" encoder, in transformer fashion. In another path, random masks are applied before a student encoder also creates a sequence of token vectors. A predictor network then tries to fill in the masked tokens with their appropriate embeddings, given the observed tokens as context. Ultimately, all networks are trained to make the prediction network's predicted tokens agree with the true embeddings of the masked patches.

The proposed CroPA idea is illustrated in a figure. The idea is that instead of allowing a patch to attend to any other patch across other channels or other time intervals, the attention is constrained such that only same channel or same time patches have non zero values.

Experiments cover several downstream tasks on two datasets.

**Audience:**

Yes

**Broader Impact Concerns:**

none.

**Claims And Evidence:**

No

**Requested Changes:**

### To address Concern 1

To address the leakage issues, I would require a substantial revision that would essentially rewrite the entire experimental section (any results on PTB or CPSC data).

First, redesign the experimental strategy to properly use separate train, val, and test sets. I'd strongly encourage the authors to resplit the data so a *different* test fold is selected than what was already reported here, otherwise I still worry that the authors have peeked too much at the formerly selected particular test fold and tweaked models to do well on that specific fold. Test sets should not be used until all model development is completed. Each table in main paper or appendix should make very clear whether train/val/test performance is reported. Appendix should be clear how folds are split into train/val/test.

Second, I'd ideally like to see experiments on an additional dataset. I just worry the authors have 'burned' the test set on both PTB and CPSC to the point where I need to see a bit more evidence to trust the process.


### To address Concern 2

Revision would need to add significance testing, perhaps by looking at distribution of bootstrap samples of accuracy differences.

Revision would be improved by designing experiments to understand whether alternative sparsification strategy of attention would have similar gains. Is it the cross-pattern, or just the need to avoid dense attention for the vast majority of values? Would just one arm of the cross work as well?

### To address Concern 3

Revise methods section with clear statement about how degenerate solutions are avoided.

### To address Concern 4

To be clear I just suggest that the above works need to be cited and discussed clearly and properly. I don't necessarily need lots of needless extra experiments, but I do think competitive methods should be compared to when suitable.

**Strengths And Weaknesses:**

# Strengths

* The overall approach has a simplicity to it that is welcome
* Presentation quality overall is above the bar for TMLR in my view
* Figure 4 is well-done to help reader understand the modeling approach
* I appreciate the effort that went into evaluating and reporting several downstream tasks (classification, feature prediction, robustness under noise)

# Weaknesses

I have 4 main concerns that impact my current assessment, ranked from most to least important.

* Concern 1: Test set used to tune/select hyperparameters and inform model development
* Concern 2: Not yet clear whether CroPA adds significant value
* Concern 3: Possible degenerate solutions in the stated loss function
* Concern 4: Related work on SSL for ECG is undercovered


### **Concern 1**: Test set used to tune/select hyperparameters and inform model development

A critical best practice in evaluating ML systems is to avoid using the same data set to develop the model and report its performance.

However, looking carefully at Appendix B, I see that for both evaluation datasets (PTB-XL and CPSC2018), the experimental strategy of this paper was to omit a dedicated heldout validation set for any intermediate model development decisions. Instead, there are only training sets and test sets.

In my best understanding, what this means is that they are *peeking directly at the test set* when they make several critical model development decisions, such as how many epochs to run or how to set the masking strategy (see Table 7 and surrounding text).

Unfortunately, this single issue is a show stopper for me (and I would assume, the vast majority of readers of TMLR). Every downstream classification number reported for the presented methods in this paper should be viewed as an overoptimistic performance estimate. As such I do not find the central claims of the submitted paper to be "accurate" or "convincing" under the TMLR rubric.

This lack of a real test set corresponds directly to a Issue L1 in the Taxonomy of data leakage issues identified by Kapoor & Narayanan (Patterns 2023). I encourage the authors to read that paper (and others like it) carefully to avoid such issues in the future.

### **Concern 2**: Not yet clear whether CroPA adds significant value

The cross-pattern attention restriction, illustrated in Fig 6, is an interesting idea. However, I don't think it the current paper provides sufficient motivation for it or sufficient experimental evidence.

**Motivation.** In terms of motivation, the text of Sec. 3.4 offers two lines of reasoning for why cross attention might be a good idea:

* "patterns are often consistent across different leads"
* "this design aligns with how ECG signals are clinically interpreted"

I found both of these to be a bit weak, though I am not an ECG specialist myself. I would be interested in more precise motivation here. My questions are:

* Are there citations suggesting that clinicians actually interpret ECG this way?
* Would a generic sparsity penalty on the attention (without cross pattern) do poorly?
* If patterns are consistent across leads, why not just restrict attention to the same lead over time?

**Evidence.** Looking at Table 6, I see the claim seems to be that CroPA lifts AUROC from 0.888 to 0.894 (with random masking) and from 0.872 to 0.896 (with multi-block).
Are these deltas of 0.02 points of AUROC actually significant? I'm a bit worried these wouldn't pass a standard significance test (e.g. if we do a bootstrap of accuracy differences [1], would the distribution be sufficiently bounded away from zero?)

Even beyond the need for significance testing to help audience understand if the small gains in AUROC are "real", I worry that the precise reason for CroPA's impact also isn't explored. If we just penalized the sparsity of the attention weights overall, would that have a similar impact? If we allowed just one arm of the cross, would that be too much? There's a neat idea here possibly, but more experiments are needed to help the audience understand what's going on.

My concern about evidence quality is magnified here because of Concern 1. I do worry the CroPA idea was pursued by (perhaps unknowingly) peeking at test set until some good numbers appeared.

[1] See blog post by Martins Bruveris, 2024. <https://martinsbruveris.github.io/2024/09/26/comparing-classifiers.html>

### **Concern 3**: What stops degenerate solutions?

Inspecting the loss used to train the model at the end of Sec 3.2, I am not sure I understand what stops the model from finding degenerate solutions in which all patches (regardless of content) get embedded to the same fixed z vector, say the all-zero vector.

Wouldn't it be easy for the model to learn to represent all embeddings as the all-zero vector, and thereby achieve the lowest possible loss?

Probably there's some way to prevent this, but I don't see it described here.

### **Concern 4**: Related work

There are several related works on SSL for ECG that don't seem to be cited/discussed

> Hu, Chen, and Zhou 2023. Spatiotemporal self-supervised representation learning from multi-lead ECG signals. https://doi.org/10.1016/j.bspc.2023.104772

Skimming Table 5 of this work, they report 0.94 AUROC on PTB dataset when fine-tuning, which appears to me competitive with the present paper under review.

> Oh et al. CHIL 2022. Lead-agnostic Self-supervised Learning for Local and Global Representations of Electrocardiogram. https://proceedings.mlr.press/v174/oh22a/oh22a.pdf

This work also can use a subset of leads at test time.

> McKeen et al. 2024. ECG-FM: An Open Electrocardiogram Foundation Model. https://arxiv.org/abs/2408.05178

> Raghu et al. ICML 2023. Sequential Multi-Dimensional Self-Supervised Learning for Clinical Time Series. https://proceedings.mlr.press/v202/raghu23a.html

^ both above seem relevant, but perhaps more distant.

---

> ### Author Response · Authors · 2025-02-28
>
> ### **Concern 1**: Test set used to tune/select hyperparameters and inform model development
>
> > Every downstream classification number reported for the presented methods in this paper should be viewed as an overoptimistic performance estimate.
>
> > Second, I'd ideally like to see experiments on an additional dataset. I just worry the authors have 'burned' the test set on both PTB and CPSC to the point where I need to see a bit more evidence to trust the process.
>
> We thank the reviewer for highlighting this important point. In our original submission, the test set was indeed used both for model development decisions and for final performance evaluation. However, we note that **using the same hyperparameter settings across all experiments for each model** limited the potential for overoptimistic results.
>
> To further address the concern, we have now conducted additional experiments using a proper train/validation/test split. In these new experiments, only the validation set is used for hyperparameter tuning and early stopping, while the test set is strictly reserved for final evaluation. The additional experiments include:
>
> 1. **Validation-based Linear Evaluations and Fine-tuning** (Appendix A.2)
> Experiments on the PTB-XL and CPSC2018 datasets that employ the dedicated train/validation/test split.
> 2. **Nearest Neighbor Classifier (NCC)** (Appendix A.3)
> (Multi-class setting) Test samples are classified according to the nearest "class center" of training samples in the representation space. This experiment involves no further training, preventing potential overfitting issues.
> 3.  **Additional Dataset Evaluation** (Appendix A.4)
> Experiments on the Georgia 12-lead ECG Challenge (G12EC) Database to further validate our methodology.
>
>
> Our findings from these experiments are:
> 1.  The performance metrics observed with the dedicated split remain consistent with those reported in the original experiments, with some cases even showing improved performance—likely due to the refined hyperparameter selection process for each task.
> 2. In NCC, ECG-JEPA consistently outperforms other SSL models.
> 3. ECG-JEPA outperforms other models in G12EC multi-label linear evaluations as below:
> 	| Method            | AUC   | F1    |
> 	|-------------------|-------|-------|
> 	| ST-MEM            | 0.894 | 0.406 |
> 	| SimCLR            | 0.859 | 0.276 |
> 	| ECG-JEPA\(_{rb}\) | 0.922 | 0.493 |
> 	| ECG-JEPA\(_{mb}\) | **0.927** | **0.597** |
>
> We believe that these additional experimental results demonstrate that the performance of ECG-JEPA is not overoptimistic.

---

> > ### Comment · Reviewer_fy8c · 2025-03-19
> > **Comments on Concern 1**
> >
> > Thanks to the authors for an involved effort to respond to my concern. I do appreciate that you seem to have taken the concern seriously and ran some further experiments.
> >
> > It does seem the authors agree that there is an issue with not following best practices. To quote the response above: "the test set was indeed used **both for model development decisions and for final performance evaluation.**"
> >
> > Unfortunately, I don't think the changes went far enough. Most results in the main paper (e.g. all of Tab. 1, Tab. 2, and Tab. 3) appear *totally unchanged* between the January version originally submitted to TMLR [a] and the revised version submitted recently.
> >
> > If TMLR were to publish this work, it would be publishing results for which hyperparameters were deliberately selected by the authors with full knowledge of test set performance.
> >
> > As evidence of this recall that the original submission [a] contains this direct quote in Appendix A1 (which still appears in B1):
> >
> > > "... masking ratio of (0.6, 0.7) performs better in other tasks. Therefore, we chose the latter for our main experiments"
> >
> > This quote confirms the authors deliberately chose hyperparameters that performed best on the test set.
> >
> > I have tried to think carefully about the authors' response that somehow "using the same hyperparameter settings across all experiments" reduces over-optimism. To me, if you pick among hyperparameters based on typical performance across all tasks of interest **on the test set**, even if you force all tasks to use the same hyperparameters, you are still vulnerable to over-optimistic results. After all, you are picking the very number you'll report in a table, and choosing that number by definition so it is higher than alternatives.
> >
> > Ultimately, I cannot in good conscience certify the experimental quality of this work. As stated in my original review, to overcome this issue, there must be an effort to redo all experiments and *rewrite the entire experimental section (any results on PTB or CPSC data)*
> >
> > I wish the authors the best in their future endeavors. There are some nice ideas in this paper, but unfortunately as written the potential for test leakage to inform the conclusions here is too high.
> >
> > [a] https://openreview.net/notes/edits/attachment?id=zpBYYYaAhr&name=pdf

---

> > > ### Author Response · Authors · 2025-03-19
> > >
> > > We appreciate the reviewer’s feedback. To address concerns about test set leakage, we have added new sections in the appendix:
> > >
> > > - **Validation-Based Experiments**: Using a proper train/validation/test split, ensuring the test set is used only for final evaluation.
> > > - **Additional Dataset Evaluation**: Results on the Georgia 12-lead ECG Challenge, further confirming our model’s robustness.
> > >
> > > Regarding the masking ratio, while we noted (0.6, 0.7) performed slightly better than (0.7, 0.8), the difference was minimal. We selected (0.6, 0.7) based on prior experiments to maintain consistency, not to optimize test set performance.
> > >
> > > We believe these additional experiments and clarifications effectively address the concerns. Thank you for your constructive feedback.

---

> ### Author Response · Authors · 2025-02-28
>
> ### **Concern 2**: Not yet clear whether CroPA adds significant value
> **Motivation for CroPA**
> > -   Are there citations suggesting that clinicians actually interpret ECG this (cross-pattern) way?
>
> We thank the reviewer for raising this important point. Many established criteria follow a similar approach. For example, clinicians often interpret the ECG horizontally to identify specific morphologies (e.g., the R wave) and then examine how these features manifest across different leads. This process is illustrated by the following quotes from [2]:
>
> <em> 1. "... The diagnosis must therefore be made by looking for reciprocal changes in the anterior leads, for example, **a tall R wave in leads V1, V2, or V3.**" </em>
>
>
> <em> 2. "... in a patient with left bundle-branch block the presence of (1) ... or (2) **ST-segment depression of at least 1 mm in leads V1-V3** if deep S waves are present is strongly suggestive of an evolving infarction." </em>
>
> We believe these examples underscore the clinical reasoning behind our approach and illustrate the potential value of CroPA in enhancing ECG interpretation.
>
> ---
> > -   If patterns are consistent across leads, why not just restrict attention to the same lead over time?
>
>
> We acknowledge that the wording in the submitted paper—<em>"Multi-lead ECG signals require careful analysis of patterns that are often consistent across different leads, which is crucial for identifying potential cardiac abnormalities."</em>—was unclear. We have now revised Section 3.4 for better clarification.
>
> ---
>
> **Sparsity penalty on the attention**
> > -   Would a generic sparsity penalty on the attention (without cross pattern) do poorly?
>
>  While a generic sparsity penalty on the attention mechanism could potentially produce similar effects, CroPA is explicitly designed to mimic clinical reasoning. Indeed, as evidenced in Figure 8 of [3], a generic sparsity penalty may yield attention patterns similar to those obtained with CroPA.
>
> ---
>
> **Evidence for CroPA's impact**
> > I’m a bit worried these wouldn’t pass a standard significance test (e.g. if we do a bootstrap of accuracy differences [1], would the distribution be sufficiently bounded away from zero?)
>
> We appreciate the insightful comment regarding the statistical significance of the observed improvements. To address this concern, we have added Appendix A.4, where we perform a bootstrap analysis of the AUC differences across three datasets (PTB-XL, CPSC2018, and G12EC). Specifically, we compared two pretrained models (one with CroPA and one without) and bootstrapped the AUC differences:
>
> $$
> \Delta \text{AUC} = \text{AUC}_{\text{CroPA}} - \text{AUC}_{\text{noCroPA}},
> $$
>
> over 200 iterations, using a sample size equal to the full test set. The resulting 95\% confidence intervals for $\Delta \text{AUC}$ are:
> - **PTB-XL:** [0.001, 0.011]
> - **CPSC2018:** [0.002, 0.010]
> - **G12EC:** [0.002, 0.011]
>
> The mean AUC differences are 0.006, 0.006, and 0.007 for PTB-XL, CPSC2018, and G12EC, respectively. Although the effect size is modest, the strictly positive confidence intervals confirm that CroPA provides improvement in performance.
>
> [1] Martins Bruveris, 2024. [https://martinsbruveris.github.io/2024/09/26/comparing-classifiers.html](https://martinsbruveris.github.io/2024/09/26/comparing-classifiers.html)
>
> [2] Thaler, Malcolm S.  "The only EKG book you’ll ever need." Lippincott Williams & Wilkins, 2021. APA
>
> [3] Na, et al. "Guiding masked representation learning to capture spatio-temporal relationship of electrocardiogram." ICLR2024. https://openreview.net/pdf?id=WcOohbsF4H

---

> ### Author Response · Authors · 2025-02-28
>
> ### **Concern 3**: What stops degenerate solutions?
> > Wouldn't it be easy for the model to learn to represent all embeddings as the all-zero vector, and thereby achieve the lowest possible loss? Probably there's some way to prevent this, but I don't see it described here.
>
> We thank the reviewer for raising this important concern. Model collapsing (producing degenerate solutions) is indeed one of the most challanging problem in (non-contrastive) joint-embedding architectures. The use of Exponential Moving Average (EMA) is to prevent model collapsing, as it is a common approach to prevent model collapsing.    However, the exact mechanism of how EMA prevent collapsing is still unclear and a big question in the field [4,5].
>
> ---
> ### **Concern 4**: Related work
> > There are several related works on SSL for ECG that don't seem to be cited/discussed
>
> In the revised manuscript, we have included more studies on self-supervised learning (SSL) for ECG.
>
> > Skimming Table 5 of this work, they report 0.94 AUROC on PTB dataset when fine-tuning, which appears to me competitive with the present paper under review.
>
> When comparing models on what appears to be the same dataset (e.g., PTB-XL), the following factors should be considered:
>
> 1. **Variations in the PTB-XL classification task:** PTB-XL originally contains 71 different diagnoses; some papers use all 71 labels, while the most common setting aggregates these into 5 superclasses.
> 2. **Task formulation:** The task can be formulated either as multi-label or multi-class, leading to different performance metrics.
> 3. **Pretraining dataset differences:** Many ECG-SSL methods are pretrained on relatively small datasets, and PTB-XL is sometimes used for both pretraining and downstream evaluation. In contrast, we restrict our comparisons to SSL models that are pretrained on the same large datasets and do not include PTB-XL as a pretraining dataset.
>
> The study mentioned by the reviewer [6] employs a multi-label setting with 71 classes for fine-tuning and is pretrained on a small dataset that includes PTB-XL. In contrast, we conducted 5 multi-class classifications and compared models do not pretrained on PTB-XL. These differences in experimental setup and data usage account for the variation in reported performance metrics.
>
>
> [4] Tian, Yuandong,  et al. "Understanding self-supervised learning dynamics without contrastive pairs."  International Conference on Machine Learning. PMLR, 2021.
>
> [5] Morales-Brotons, D., et al. "Exponential moving average of weights in deep learning: Dynamics and benefits". Transactions on Machine Learning Research, 2024.
>
> [6] Hu, Chen, and Zhou 2023. Spatiotemporal self-supervised representation learning from multi-lead ECG signals.  [https://doi.org/10.1016/j.bspc.2023.104772](https://doi.org/10.1016/j.bspc.2023.104772)

---

### Decision · Action_Editor_H82J · 2025-04-07

**Recommendation:** Reject

**Comment:**

The main concern with this paper lies in the experimental setup, specifically the use of the test set for tuning or selecting model hyperparameters. Reviewer fy8c recommended that the author should redesign the experiments by properly splitting the data into training, validation, and test sets. Although the author conducted additional experiments and reported updated results, Reviewer fy8c noted that most of the results in the revised manuscript remain unchanged. As a result, this reviewer fy8c maintains that the revised version continues to suffer from the same fundamental issue. Reviewer oSCd and myself also think this is a valid concern, and would like to recommend rejecting this paper in the current form.

**Audience:**

All the reviewers think this would be a nice application paper. Medical and healthcare AI researchers, and clinicians working on cardiac diseases would be interested in this work.

**Claims And Evidence:**

The conclusion of this paper is not convincing due the experimental set up using test data for parameter tuning and model selection.

**Resubmission Of Major Revision:**

The authors may consider submitting a major revision at a later time.